# Competition between microtubule-associated proteins directs motor transport

Brigette Y. Monroy[1], Danielle L. Sawyer[1], Bryce E. Ackermann[1], Melissa M. Borden[1], Tracy C. Tan[1] & Kassandra M. Ori-McKenney[1]

Within cells, motor and non-motor microtubule-associated proteins (MAPs) simultaneously converge on the microtubule. How the binding activities of non-motor MAPs are coordinated and how they contribute to the balance and distribution of motor transport is unknown. Here, we examine the relationship between MAP7 and tau owing to their antagonistic roles in vivo. We find that MAP7 and tau compete for binding to microtubules, and determine a mechanism by which MAP7 displaces tau from the lattice. MAP7 promotes kinesin-based transport in vivo and strongly recruits kinesin-1 to the microtubule in vitro, providing evidence for direct enhancement of motor motility by a MAP. Both MAP7 and tau strongly inhibit kinesin-3 and have no effect on cytoplasmic dynein, demonstrating that MAPs differentially control distinct classes of motors. Overall, these results reveal a general principle for how MAP competition dictates access to the microtubule to determine the correct distribution and balance of motor activity.

---

[1] Department of Molecular and Cellular Biology, University of California, Davis, Davis, CA 95616, USA. Correspondence and requests for materials should be addressed to K.M.O-M. (email: kmorimckenney@ucdavis.edu)

The balance of intracellular transport is essential for the structural and functional organization of a cell. The microtubule motors, kinesin-1 and cytoplasmic dynein, drive cellular cargoes toward microtubule plus-ends and minus ends, respectively. Mutations in either motor pathway disrupt the balance of transport and cause a range of diseases[1,2]. Motors must navigate a crowded microtubule lattice that is decorated with non-motor MAPs. These MAPs have various roles in regulating microtubule dynamics, turnover, and stability, as well as in influencing molecular motor transport[3–6]. The Alzheimer's disease related MAP, tau (MAPT), is highly enriched in neuronal axons and inhibits kinesin-1 motility in vivo and in vitro, but has less of an effect on dynein-based movement[3,7–11]. How plus-end directed transport is accomplished in the tau-rich axonal environment is therefore an outstanding question, though the effects of tau on other classes of kinesin motors have not been tested. One MAP that appears to be antagonistic to tau is MAP7 (EMAP-115, ensconsin)[3,7–15]. The knockdown of MAP7 or tau in neurons produces opposite axonal branching phenotypes[12,13]. In addition, MAP7 is important for kinesin-based cargo transport and nuclear positioning in vivo[16–18]; however, the molecular mechanism underlying this relationship is unclear. How tau and MAP7 activities are coordinated on the surface of individual microtubules in cells, and the functional consequence of MAP distributions on microtubule motor motility remain open questions.

Here, we present a detailed analysis of the competition between tau and MAP7 and a molecular dissection of the functional interaction between MAP7 and kinesin-1. We find that MAP7 actively competes with and displaces tau from the microtubule, and determine a molecular mechanism by which MAP7 invades and displaces tau from the lattice. In striking contrast to the inhibitory effect of tau, MAP7 promotes kinesin-based transport in vivo and enhances kinesin-1 binding to the microtubule ~150-fold in vitro, providing evidence for direct regulation of a motor by a MAP. In addition, we find that MAP7 and tau strongly inhibit kinesin-3, but have no effect on dynein motility, suggesting that individual MAPs may provide differential control over distinct classes of microtubule motors. Our work illustrates a general principle for how competition between microtubule-associated proteins directs and distributes molecular motors to maintain a balance of transport within the crowded intracellular environment.

## Results

### MAP7 and tau exhibit overlapping localization patterns.
We first examined the localization patterns of MAP7 and tau in mature *Drosophila* peripheral nervous system neurons in vivo and DIV4 primary mouse neuronal cultures. In both systems, we found that MAP7 localized within both dendrites and axons, while tau was predominantly restricted to the axons, as previously reported (Fig. 1a–d)[13,19]. Based on their overlapping spatial patterns within neurons, we wanted to investigate whether MAP7 and tau could bind simultaneously to individual microtubules. Using total internal reflection fluorescence microscopy (TIRF-M), we imaged purified fluorescently labeled human 2N4R tau and full-length MAP7 (Supplementary Figure 1) binding to taxol-stabilized microtubules. We first mixed equimolar concentrations of both MAPs in the flow chamber and observed that tau is strongly excluded from sites of MAP7 enrichment, and vice versa (Fig. 2a–d). A fivefold excess of MAP7 completely abolished tau binding to the microtubule (Fig. 2a, b). Conversely, even at 100-fold excess of tau, MAP7 remained tenaciously bound to the microtubule in distinct patches that were devoid of tau (Fig. 2c, d). These experiments suggest that MAP7 and tau compete for binding to the microtubule surface.

### MAP7 and tau compete for binding on the microtubule lattice.
We next performed time-lapse imaging of MAP7 and tau dynamics on microtubules, and observed that while tau initially bound microtubules more rapidly than MAP7, over longer time periods, MAP7 accumulated and displaced tau from the lattice (Fig. 2e, f). There are two non-mutually exclusive possibilities for how MAP7 and tau could compete for the microtubule: (1) they share overlapping binding sites on the microtubule, or (2) domains of each MAP not involved in microtubule binding could have a role in hindering the binding of the other. We tested these possibilities by purifying a truncated version of MAP7 that contains the microtubule-binding domain, but lacks a conserved negatively charged C-terminal region (-10.7 net charge, MAP7ΔC). Using a TIRF-M-based microtubule-binding assay, we found that MAP7ΔC exhibits a slightly higher microtubule-binding affinity compared with full-length MAP7 based on the $K_D$ derived from saturation curves ($3.03 \pm 1.2$ vs. $1.39 \pm 0.5$ μM (means ± s.d.) for full-length MAP7 and MAP7ΔC, respectively; Supplementary Figure 2a–b). Although MAP7ΔC still accumulates on the microtubule in patches that exclude tau, its ability to evict tau from the microtubule is greatly reduced, indicating that MAP7 and tau do share overlapping binding sites, but the negatively charged C-terminal domain of MAP7 plays a secondary role in evicting tau molecules that are already associated with the lattice (Fig. 2g, h). Solution-based pull-down assays revealed that purified MAP7 and tau proteins do not interact with one another (Supplementary Figure 2c). In addition, we observed that the C-terminus of MAP7 (MAP7ΔN) is unable to bind to microtubules (Supplementary Figure 2d–e), or displace tau from the lattice (Supplementary Figure 2f–g).

Next, we evaluated the microtubule-binding behavior of MAP7 and tau. Microtubule co-sedimentation assays revealed that MAP7 bound microtubules with an apparent $K_D$ of $0.46 \pm 0.06$ μM, whereas tau displayed a threefold higher $K_D$ of $1.56 \pm 0.16$ μM in our conditions (means ± s.d.; Fig. 2i, j). These values are similar to previous in vitro results and suggest that MAP7 has a higher microtubule-binding affinity than tau[14,20]. Nonetheless, we observed tau binding to microtubules before MAP7 in our TIRF-M assays, which could be attributed to the non-canonical binding exhibited by tau[21,22]. Single-molecule dwell time analysis revealed that MAP7 molecules bind the microtubule ~40-times longer than tau molecules in our conditions (82.3 s vs. 1.9 s; Fig. 2k, l and Supplementary Figure 2h), though these binding dynamics are likely modulated by external factors in vivo, such as by phosphorylation[23–25]. These results indicate that MAP7 and tau compete for a common binding site on the microtubule and MAP7 is able to invade tau-rich regions of the microtubule through a mechanism that involves its negatively charged C-terminal domain, its higher microtubule-binding affinity, and longer lattice residency times. Interestingly, a previous study hypothesized that tau could be easily displaced from the microtubule owing to its weak binding interaction with the lattice[26].

### *Drosophila* ensconsin enhances kinesin transport in vivo.
To determine the functional consequence of the MAP7-tau competition, we analyzed the effects of tau and ensconsin, the *Drosophila* homolog of MAP7, overexpression in larval dendritic arborization (DA) neurons. Tau overexpression led to fewer branches, whereas ensconsin overexpression caused an increase in the number of dendrite branches and total dendrite length (Fig. 3a–c). Overexpression of both tau and ensconsin still led to an increase in the number of branches and total branch length (Fig. 3a–c), consistent with the ability of MAP7 to outcompete tau in vitro. Interestingly, the increase in proximal branching that

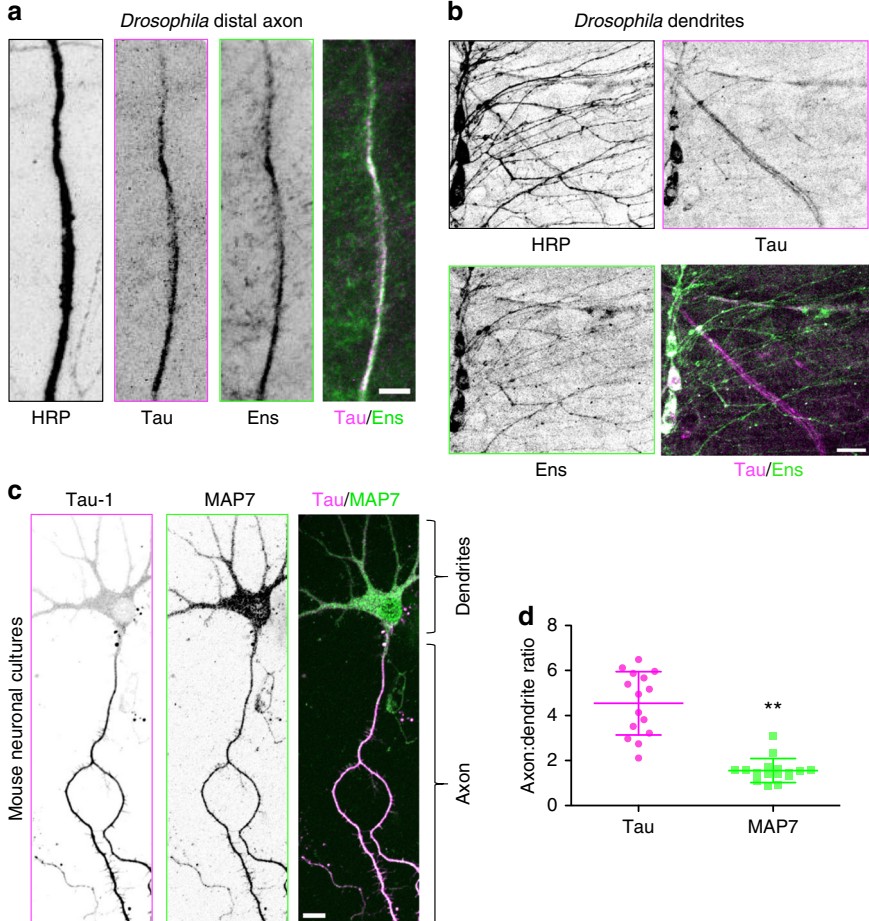

**Fig. 1** MAP7 and tau exhibit overlapping localization patterns in *Drosophila* and mammalian neurons. Immunohistochemical staining of larval fillets from an endogenous tau-GFP fly line with antibodies against HRP, GFP, and ensconsin reveals that **a** tau and ensconsin are expressed within axons, and **b** ensconsin, but not tau, is expressed in the dendrites of peripheral nervous system neurons. Scale bars are 10 and 15 μm for **a** and **b**, respectively. **c** Immunocytochemistry of mouse DIV4 neuronal cultures with antibodies against dephospho-tau (Tau-1) and MAP7. Scale bars are 20 μm. **d** Quantification of the axon to dendrite fluorescence intensity ratio for MAP7 is 1.5 ± 0.5 (mean ± s.d.), indicating a near uniform distribution of MAP7 throughout the neuron, whereas the ratio for tau is 4.5 ± 1.4 (mean ± s.d.), indicating a fivefold enrichment of tau in the axon (** indicates $P < 0.0001$ using a student's *t*-test, $n = 15$ and 16 neurons for tau and MAP7, respectively, from two independent cultures)

results from ensconsin overexpression phenocopies dynein knockdown in these neurons[27], indicating this phenotype may be the result of an imbalance in motor transport. We examined the localization pattern of Golgi outposts because this cargo is bidirectionally transported in dendrites and is important for branch formation within the DA neurons[28,29]. In control neurons, Golgi outposts are enriched within the cell body, within the primary and secondary branches, at branchpoints, and at distal tips (Fig. 3d)[30]. Overexpression of ensconsin redistributed Golgi outposts from the primary and secondary branches to the cell body and the extreme distal tips of dendrite branches (Fig. 3d, e). Within the primary branches of the DA neurons, microtubules are uniformly organized with their plus-ends toward the cell body (Fig. 3f)[30,31]. Within the terminal branches, microtubules are uniformly organized with their plus-ends toward the distal tips (Fig. 3f)[30]. Therefore, based on the orientation of microtubules within these neurons, these data indicate that overexpression of ensconsin influences the balance of transport in favor of kinesin, driving plus-end directed movement of kinesin cargoes such as Golgi outposts. Accumulation of Golgi outposts near the cell body may facilitate nascent branch formation in the proximal region via microtubule nucleation[30,32], providing an explanation for the phenotype resulting from ensconsin overexpression.

**MAP7 directly recruits kinesin-1 to the microtubule**. Kinesin-1 has been reported to drive the anterograde transport of intracellular cargoes within the dendrites of the DA neurons[33], indicating the overexpression of ensconsin could indeed enhance kinesin-1 transport. In addition, previous studies have reported that MAP7 is necessary for kinesin-1 activities in vivo[16–18,34]; however, the molecular mechanism by which MAP7 affects kinesin-1, either directly or indirectly, is unknown. To determine whether MAP7 directly affects kinesin-1, we analyzed the dynamics of purified MAP7 and K560, a well-characterized truncated version of human kinesin-1[35], on taxol-stabilized microtubules using TIRF-M. We found that the presence of saturating MAP7 on the microtubule (Supplementary Figure 2a) markedly enhanced the amount of K560 on the microtubule 165-fold (Fig. 4a), and increased the landing rate of K560 ~15-fold (from $0.09 \pm 0.11$ to $1.34 \pm 0.43$ motors μm$^{-1}$ min$^{-1}$. Fig 4b). MAP7 also slightly increased the processivity (from 873.8 ± 43.3 to 984.3 ± 87.5 nm) and decreased the velocity of K560 (from 434.0 ± 111.2 to 327.5 ± 128.5 nm/sec, Fig. 4c, d) in our assays. To test the effects of MAP7 on K560 enzymatic activity, we measured the basal and microtubule-stimulated ATPase activities of K560 in the presence and absence of MAP7. The basal ATPase activity of K560 was unchanged by MAP7; however, MAP7 lowered the

$K_{mMT}$ of K560 from 1.46 ± 0.21 μM to 0.27 ± 0.04 μM, indicating that MAP7 increases the affinity of K560 for the microtubule without affecting its ATPase turnover rate (Fig. 4e).

How might MAP7 enhance kinesin-1 binding to microtubules? MAP7 could recruit kinesin-1 to the microtubule and remain stably associated with the motor as it moves, acting as a tether, or MAP7 could recruit kinesin-1 through a transient interaction, releasing the motor once it begins to step along the microtubule. In addition, MAP7 could indirectly recruit the motor by altering the microtubule lattice to favor motor binding. Dual-color single-

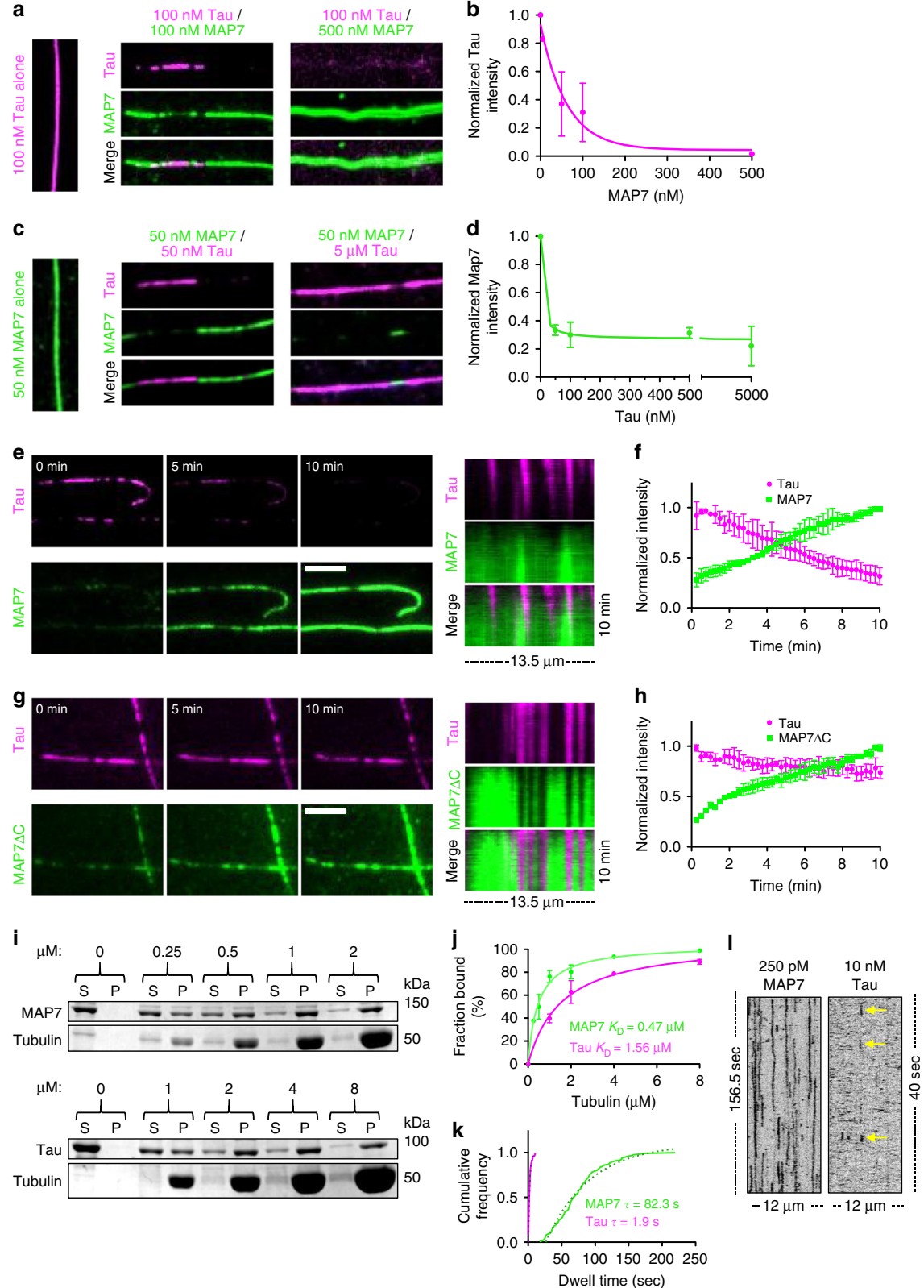

molecule experiments revealed that < 1% of K560 molecules co-migrated with a MAP7 molecule (3/328 K560 molecules; Fig. 4f) under our conditions. This result suggests a recruitment, rather than a tethering role for MAP7 in enhancing kinesin-1 microtubule association.

We next examined if kinesin-1 directly interacts with MAP7. Solution-based pull-down assays revealed that both MAP7 and ensconsin modestly interact with K560 (Supplementary Figure 3a–b), consistent with interaction studies done in cell lysates[17]. We turned to *Drosophila* ensconsin to refine the previously reported kinesin-1-binding domain, which spans 158 amino acids and contains a predicted coiled-coil domain[17]. Similar to human MAP7, *Drosophila* ensconsin also strongly recruited K560 to the microtubule (Fig. 5a, b). However, there is relatively low amino-acid homology between MAP7 and ensconsin within this region (41% identity). Based on sequence conservation, we deleted aa 665–699 within ensconsin (Supplementary Figure 3c). Removal of these amino acids did not affect microtubule association, but strongly perturbed the ability of ensconsin to recruit K560 to the microtubule (Fig. 5a, b). Excising the most conserved coiled-coil heptad within this region (693–699, + 4 charge) also strongly disrupted ensconsin's recruitment of K560 to the microtubule (Fig. 5a, b). Using truncation mutants of kinesin-1, we found that MAP7 did not recruit K490, modestly recruited K508 (11-fold), and recruited K523 similarly to K560 to microtubules (152-fold vs. 165-fold, Fig. 5c, d, Supplementary Figure 3d). This region (500–523) has −5 charge and is conserved among eukaryotes, but is not conserved in *Aspergillus nidulans*, which lacks MAP7 (Supplementary Figure 3e). The lack of effect on K490 suggests that MAP7 does not allosterically change the microtubule lattice to favor kinesin binding, indicating MAP7 must directly interact with kinesin-1 to facilitate the interaction between the kinesin-1 and the microtubule. Overall, our results suggest that MAP7 recruits kinesin-1 to the microtubule through a transient, ionic interaction between coiled-coil domains, but does not remain stably bound to kinesin-1 as it translocates along the microtubule, consistent with a previously proposed hypothesis[16].

Having established that MAP7 and tau compete for microtubule binding and have differential effects on kinesin-1 transport, we examined how the presence of MAP7 influenced the behavior of kinesin-1 upon encountering a tau patch. We observed a similar percentage of K560 motors that detached, paused, or passed tau patches in the absence compared with the presence of MAP7 (Supplementary Figure 4a–c). However, when we performed time-lapse imaging of MAP7, tau, and K560 over the course of 10 min, we observed that although tau patches initially preclude the binding of K560 to the microtubule, over time MAP7 displaced tau and strongly recruited K560 to the same sites on the lattice (Fig. 5e). In addition, an excess of tau inhibited the microtubule-stimulated ATPase activity of K560, but the presence of MAP7 restored K560 ATPase activity ($24.7 \pm 4.0$, $2.9 \pm 0.8$, and $21.9 \pm 3.7 \, s^{-1}$ for K560 alone, K560 + tau, and K560 + tau + MAP7, respectively; Fig. 5f). This result further supports a primarily transient role for MAP7 in the recruitment of kinesin-1 to the lattice, as opposed to a mechanism whereby MAP7 tethers kinesin-1 to the microtubule through a stable interaction.

**MAP7 And Tau differentially affect other microtubule motors.** We next investigated if MAP7 affected two other classes of processive transport microtubule motors: the dendritic and axonal cargo motor, kinesin-3[36], and the ubiquitous minus-end directed dynein–dynactin motor complex (DDB)[37]. Single-molecule assays revealed that, in contrast to kinesin-1, MAP7 substantially reduced the overall amount of kinesin-3 (KIF1A) on the microtubule by fourfold and the number of kinesin-3 landing events on the microtubule approximately five-fold (from $1.50 \pm 0.74$ to $0.38 \pm 0.31$ motors/μm/min. Fig 6a–c). Tau also strongly inhibited kinesin-3 landing rate ($0.24 \pm 0.24$ motors/μm/min) and motility similarly to its effects on kinesin-1 (Fig. 6b–d, Supplementary Figure 4c). Interestingly, MAP7 had little effect on the overall amount of DDB on the microtubule or the velocity of DDB complexes ($606.4 \pm 178.0$ and $555.2 \pm 186.5$ nm/sec for DDB alone and DDB + MAP7, respectively; Fig. 6e–g). Together, these results provide a framework for how MAPs dictate access to the microtubule and subsequently determine the spatial distribution of microtubule motors within a crowded intracellular environment.

**Discussion**

Overall, our study presents a detailed analysis of the competition between tau and MAP7 and a molecular dissection of the functional interaction between MAP7 and kinesin-1. Previous studies have shown that tau inhibits kinesin-1 without markedly affecting dynein[9,10], posing a transport problem within the axons of neuronal cells (Fig. 6h). We have found that MAP7 can facilitate kinesin-1 motility in two ways without significantly affecting dynein. First, MAP7 actively competes with and displaces tau from the microtubule, and second, MAP7 directly recruits kinesin-1 to the microtubule (Fig. 6h). It has been proposed that MAP7 relieves the autoinhibition of kinesin-1 in vivo[16]. Whether

---

**Fig. 2** MAP7 and tau compete for binding on the microtubule lattice. **a** TIRF-M images of mTagBFP-tau on microtubules in the absence and presence of increasing concentrations of sfGFP-MAP7. **b** Quantification of tau fluorescence intensity as a function of MAP7 concentration ($n = 320$ microtubules from two independent trials). **c** Images of sfGFP-MAP7 on microtubules in the absence and presence of increasing concentrations of mTagBFP-tau. **d** Quantification of MAP7 fluorescence intensity as a function of tau concentration ($n = 228$ microtubules from three independent trials). Images in **a** and **c** are 10.3 μm in length. **e** Movie frames and corresponding kymographs of 50 nM mTagBFP-tau and 50 nM sfGFP-MAP7 on microtubules over 10 min. Scale bar is 5 μm. **f** Quantification of tau and MAP7 fluorescence intensity over 10 min ($n = 3$ microtubules quantified from two independent trials). **g** Movie frames and corresponding kymographs of 50 nM mTagBFP-tau and 50 nM sfGFP-MAP7ΔC on microtubules over 10 min. Scale bar is 5 μm. **h** Quantification of tau and MAP7ΔC fluorescence intensity over 10 min ($n = 3$ microtubules quantified from two independent trials). **i** Coomassie Blue-stained SDS-PAGE shows the binding behavior of 500 nM MAP7 or 500 nM tau in the presence of increasing concentrations of microtubules. **j** Results from three experiments were plotted and fit to a Michaelis–Menten equation ($K_D \pm$ s.d is $0.47 \pm 0.06$ μM for MAP7 and $1.56 \pm 0.16$ μM for tau ($P = 0.0004$)). **k** Graph depicting the cumulative frequency of MAP7 (decay constant, $\tau = 82.3 \pm 11.2$ s; $n = 206$ events from $n = 39$ microtubules from two independent trials) and tau (decay constant, $\tau = 1.9 \pm 0.1$ s; $n = 142$ events from $n = 16$ microtubules from two independent trials) dwell times fit to a one phase exponential decay ($R^2 = 0.980$ and $0.995$ for MAP7 and tau, respectively; $P < 0.0001$). **l** Corresponding kymographs for 250 pM sfGFP-MAP7 and 10 nM sfGFP-tau dwell times on microtubules (frame rates are 2 f/s and 7.8 f/s for MAP7 and tau, respectively). Yellow arrows indicate tau dwells. Means ± s.d. are plotted for all graphs. All experiments produced similar results with sfGFP-tau and TagRFP-MAP7 and were repeated with at least two separate protein preparations

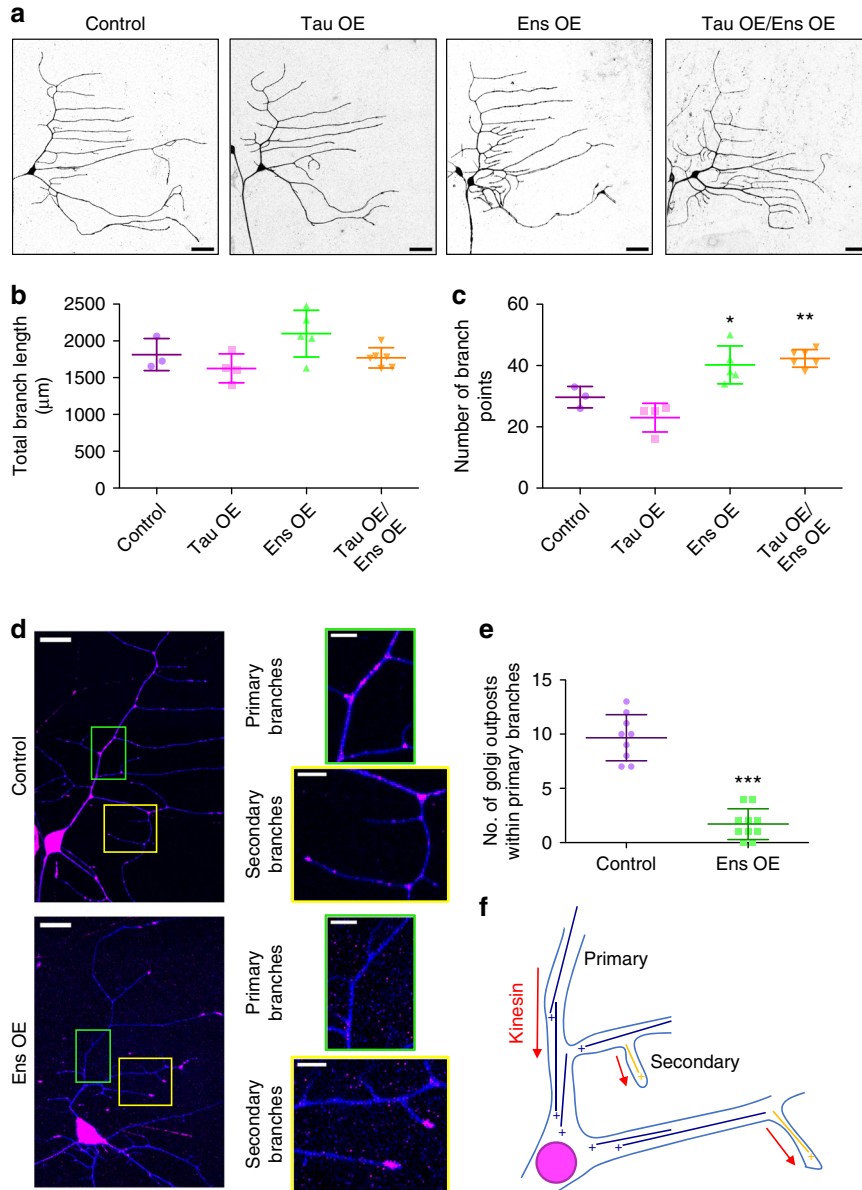

Fig. 3 *Drosophila* ensconsin enhances kinesin transport in vivo. **a** Dendrite morphologies of control, *221-Gal4* > UAS-*tau* overexpression (OE), *221-Gal4* > UAS-*ens* OE, and *221-Gal4* > UAS-*tau* OE/UAS-*ens* OE class I DA neurons. Scale bars are 30 μm. **b** Quantification of the total neuronal branch length for each genotype (means ± s.d. are 1814 ± 218, 1703 ± 151, 2099 ± 317, and 1769 ± 137 μm for control, tau OE, ens OE, and tau OE/ens OE, respectively; control vs. tau OE: $P = 0.5$; control vs. ens OE: $P = 0.2$; control vs. tau OE/ens OE: $P = 0.7$; tau OE vs. ens OE: $P = 0.09$). **c** Quantification of the number of dendritic branchpoints for each genotype (means ± s.d. are 29.7 ± 3.5, 25.3 ± 0.6, 40.2 ± 6.2, and 42.3 ± 2.9 branchpoints for control, tau OE, ens OE, and tau OE/ens OE, respectively; control vs. tau OE: $P = 0.1$; control vs. ens OE: $P = 0.03$ (*); control vs. tau OE/ens OE: $P = 0.0006$ (**); tau OE vs. ens OE: $P = 0.007$). $n = 3$, 3, 5, and 6 neurons for control, tau OE, ens OE, and tau OE/ens OE, respectively, for **b** and **c**. **d** Golgi outpost (pink) localization throughout the dendritic arbor (blue) of control and *221-Gal4* > UAS-*ens* OE class I DA neurons. Zoomed regions of primary and secondary branches are shown. Scale bars are 15 μm for full neurons and 5 μm for zoomed images. **e** Quantification of the number of Golgi outposts per 50 μm of primary branch in control vs. ensconsin OE neurons (means ± s.d. are 9.7 ± 2.1 vs. 1.7 ± 1.4 Golgi outposts for control and ensconsin OE, respectively; $P < 0.0001$ (***); $n = 10$ neurons for each genotype). Graphs in **b**, **c**, and **e** are scatterplots of all datapoints with the lines indicating the means ± s.d. A student's *t*-test was used for all statistical analyses. **f** Schematic of the orientation of microtubules within the class I DA neurons, indicating that sites of Golgi outpost localization in the ensconsin OE neurons are final destinations for microtubule plus-end transport

MAP7 has a secondary role in relieving the autoinhibition of kinesin-1 either in solution or after recruitment to the microtubule will be an interesting line of future investigation. We further show that both MAP7 and tau inhibit kinesin-3, which transports cargo into both dendrites and axons (Fig. 6h). The differential effects of these MAPs on microtubule motors not only establishes the balance of kinesin-1 and dynein transport in the

axon, but may also help to direct kinesin-3 motors into the dendrites during specific developmental stages.

An imbalance of microtubule-based transport has been implicated in disease[1,2], and our data suggest that competition between MAPs likely has a role in coordinating the correct spatiotemporal activation of particular types of microtubule motors. In neurodegenerative diseases such as Alzheimer's and fronto-

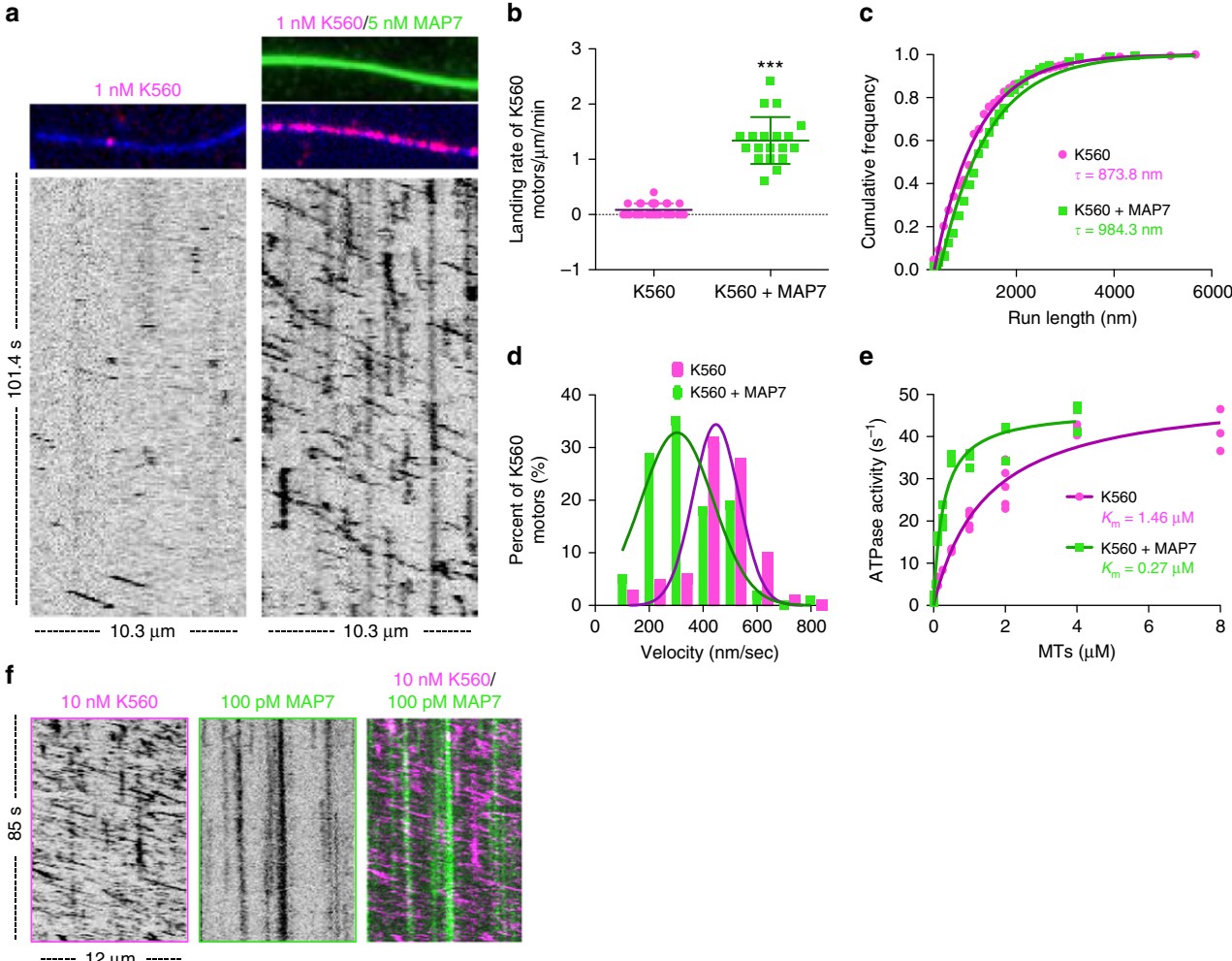

**Fig. 4** MAP7 directly recruits kinesin-1 to the microtubule. **a** TIRF-M images and corresponding kymographs of 1 nM K560-mScarlet (pink) + 1 mM ATP in the absence and presence of 5 nM sfGFP-MAP7 (green). Images are 10.3 μm in length. **b** Quantification of the landing rates of 1 nM K560-mScarlet + 1 mM ATP in the absence and presence of 5 nM sfGFP-MAP7 (means ± s.d. are 0.09 ± 0.11 vs. 1.34 ± 0.43 motors μm$^{-1}$ min$^{-1}$ for K560 alone and K560 + MAP7, respectively; $P < 0.0001$ (***) using a student's t-test; $n = 28$ events from 12 microtubules from two independent trials for K560 alone and $n = 20$ events from eight microtubules from two independent trials for K560 + MAP7). All datapoints are plotted with lines indicating means ± s.d. **c** Cumulative frequency distribution plot of K560-mScarlet run lengths ( + 1 mM ATP) in the absence and presence of sfGFP-MAP7 fit to a one phase exponential decay. Mean decay constants ± s.d. are 873.8 ± 43.3 and 984.3 ± 87.5 nm for K560 and K560 + MAP7, respectively ($P < 0.0001$; $n = 173$ and 204 K560 motors for K560 alone and K560 + MAP7, respectively, from three independent trials). **d** Velocity histograms of K560-mScarlet + 1 mM ATP in the absence and presence of sfGFP-MAP7 with Gaussian fits. Means ± s.d. are 434.0 ± 111.2 and 327.5 ± 128.5 nm/sec for K560 alone and K560 + MAP7, respectively ($P < 0.0001$; $n = 79$ and 108 K560 motors for K560 alone and K560 + MAP7, respectively, from three independent trials). **e** ATPase activities of 50 nM K560-mScarlet + 1 mM ATP in the absence and presence of 50 nM sfGFP-MAP7 as a function of microtubule concentration. All replicates are plotted from $n = 3$ independent experiments per condition and two separate protein preps and fitted with Michaelis–Menten kinetics (mean $K_m$ ± s.d. are 1.46 ± 0.21 μM and 0.27 ± 0.04 μM for K560 alone and K560 + MAP7, respectively; $P = 0.0006$). (**f**) Kymograph analysis of 10 nM K560-mScarlet + 1 mM ATP in the presence of 100 pM sfGFP-MAP7 reveals vertical, stationary MAP7 molecules (green) that do not co-migrate with processive K560 motors (pink diagonal lines)

temporal dementia, tau becomes hyperphosphorylated and dissociates from the microtubule[38]. In the absence of tau, the presence of MAP7 on the microtubule could tip the balance of transport in favor of kinesin-1, similar to what we observe upon overexpression of MAP7 in vivo (Fig. 3). This indirect disruption of dynein-based retrograde transport could have neurodegenerative effects, as has been seen in patients with mutations in dynein pathway components[1].

Our results also raise intriguing questions about competition between other MAPs. The microtubule-binding domain of MAP7 is distinct from that of tau, MAP2, and MAP4, which contain similar tandem microtubule-binding repeats[39]. Based on our results, it is possible that tau and MAP7 also compete with other

MAPs for microtubule binding. Competition or coordination between MAP7 and dendrite-specific MAPs, such as MAP2 or DCX, may be especially important for directing kinesin-3 transport, because MAP7 is present in the dendrites, but inhibits kinesin-3 in vitro. Likewise, there may be axonal MAPs that enable kinesin-3 to drive transport in the axons in the presence of both MAP7 and tau. Additionally, there are other mechanisms of regulation that could dictate the spatiotemporal association patterns of MAPs on microtubules, such as post-translational modifications of tubulin[40] or of the MAPs themselves. Overall, our results suggest that competition between MAPs may be a general mechanism for directing motor transport in cells. Combined with the tubulin-code hypothesis, which posits that

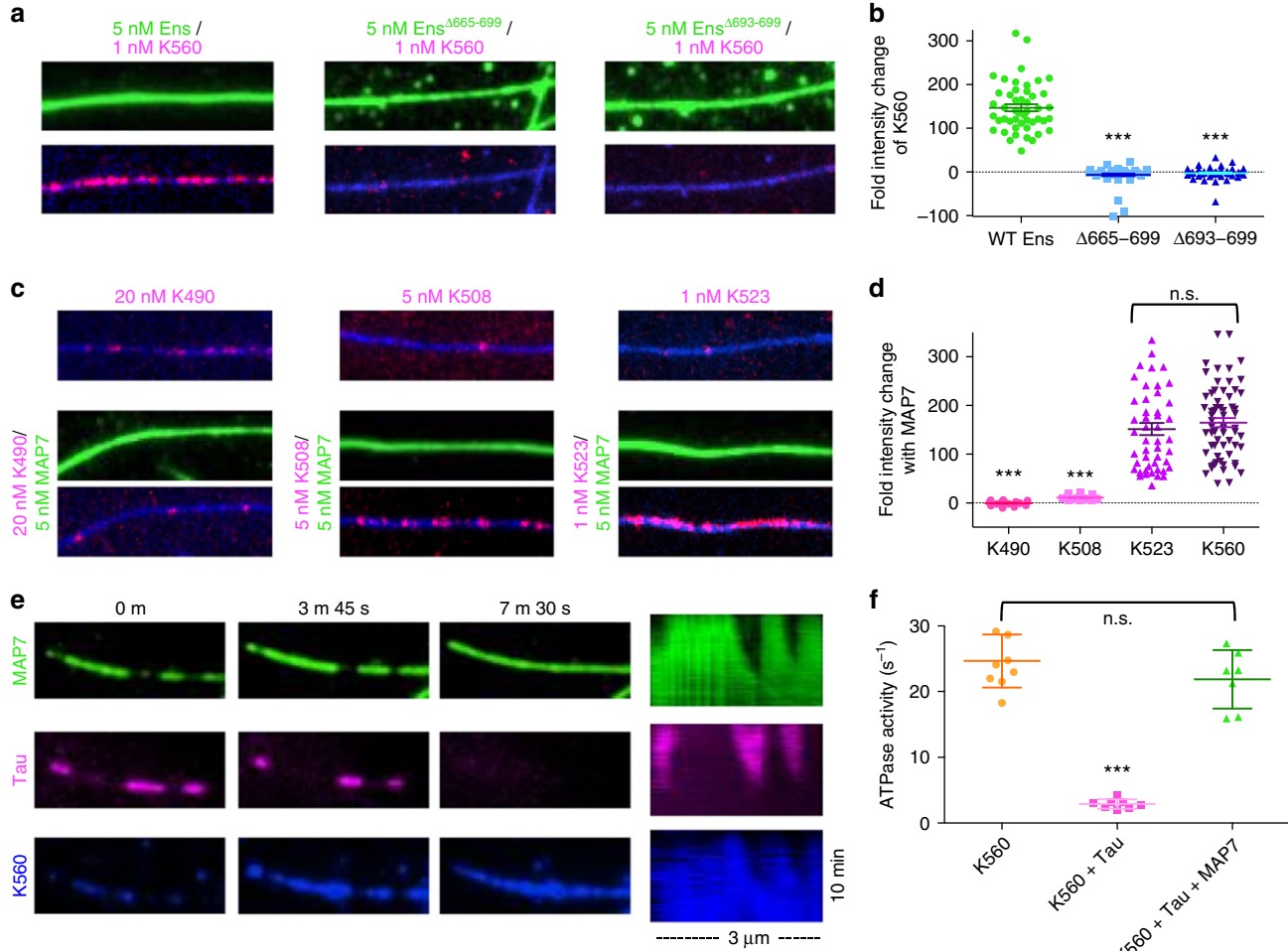

**Fig. 5** Dissection of the MAP7-kinesin-1 interaction and its role in facilitating kinesin-1 transport in the presence of inhibitory MAPs. **a** TIRF-M images of 1 nM K560-mScarlet with 5 nM sfGFP wild type or mutant ensconsin recombinant proteins. **b** Quantification of the fold change in fluorescence intensity of K560-mScarlet on the microtubule in the presence of wild type, Δ665–699, or Δ693–699 ensconsin proteins. All datapoints are plotted with lines indicating means ± s.d. (147.0 ± 56.2, −6.5 ± 20.9, and −2.5 ± 13.4-fold intensity change for wild type, Δ665–699, or Δ693–699 ensconsin, respectively; $n$ = 48, 53, and 51 microtubules, respectively from two independent trials; $P < 0.0001$ (***) for wild type vs. Δ665–699 or Δ693–699 ensconsin). **c** TIRF-M images of different kinesin-1 constructs in the absence or presence of sfGFP-MAP7. Only K523 is recruited to the same extent as K560. Images in **a** and **c** are 9.5 μm in length. **d** Quantification of the fold change in fluorescence intensity of K490-, K508-, K523-, or K560-mScarlet on the microtubule in the presence of sfGFP-MAP7. All datapoints are plotted with lines indicating means ± s.d. (−0.5 ± 4.2, 10.9 ± 4.0, 151.6 ± 86.0, and 164.9 ± 74.2 fold intensity change for K490, K508, K523, and K560, respectively; $n$ = 20, 29, 48, and 63 microtubules from two independent trials; $P < 0.0001$ (***) for K490 and K508 vs. K523 and K560). **e** Movie frames and corresponding kymographs of sfGFP-MAP7 (green), mScarlet-tau (pink), and K560–647 (blue) show MAP7 displacing tau and recruiting K560 to the microtubule lattice over 10 min. All images are 3.1 μm in length. **f** ATPase activities of 50 nM K560-mScarlet + 1 mM ATP + 2 μM microtubules in the absence and presence of 500 nM mTagBFP-tau and 500 nM sfGFP-MAP7. All replicates are plotted from $n$ = three independent experiments per condition and lines representing the means ± s.d. (24.7 ± 4.0 s⁻¹ ($n$ = 9), 2.9 ± 0.8 s⁻¹ ($n$ = 8), and 21.8 ± 3.7 s⁻¹ ($n$ = 7) for K560 alone, K560 + tau, and K560 + tau + MAP7, respectively; K560 vs. K560 + tau: $P < 0.0001$ (***); K560 + tau vs. K560 + tau + MAP7: $P < 0.0001$; K560 vs. K560 + tau + MAP7: $P = 0.1596$). All experiments were repeated with at least two separate protein preparations. A student's $t$-test was used for all statistical analyses

modifications of the tubulin subunits themselves modulate molecular motor transport[40,41], our data contributes another layer of regulation, the MAP-code, that we suggest may be similarly effectual in the spatiotemporal control of intracellular transport.

## Methods

**Molecular biology**. The cDNAs for protein expression and fly injection used in this study were as follows: human Tau-2N4R (Addgene #16316), human MAP7 (GE Dharmacon MGC Collection #BC025777), human K560 (a gift from R. Vale), *Drosophila melanogaster* Ensconsin (Drosophila Genome Resource Center-DGRC #1070880), and human KIF1A (1–393) (Addgene # 61665). Tau-2N4R, MAP7, MAP7ΔC (aa 1–353), MAP7ΔN (aa 353–749), and ensconsin proteins were cloned in frame using Gibson cloning into a pET28 or pFastBacHTA vector with an N-terminal strepII-Tag, mTagBFP, TagRFP, or a superfolder GFP (sfGFP) cassette. K560, K523, K508, K490, and KIF1A were cloned in frame using Gibson cloning into pET28 vector with a C-terminal mScarlet-strepII cassette.

**Protein expression and purification**. Tubulin was isolated from porcine brain using the high-molarity piperazine-N, N′-bis (PIPES) procedure, where tubulin is polymerized and depolymerized in repeated cycles to remove microtubule-associated proteins and other contaminants[42]. For bacterial expression of mTagBFP-Tau, sfGFP-Tau, sfGFP-MAP7, TagRFP-MAP7, sfGFP-MAP7ΔC, sfGFP-MAP7ΔN, mCardinal-MAP7ΔN, K560-mScarlet, K523-mScarlet, K508-mScarlet, K490-mScarlet, and KIF1A-mScarlet, BL21-RIPL cells were grown at 37 °C until ~O.D. 0.6 and protein expression was induced with 0.1 mM IPTG. Cells were grown overnight at 18 °C, harvested, and frozen. *D. melanogaster* sfGFP-Ensconsin proteins were grown at 20 °C for 4 h before being harvested. Cell pellets were resuspended in lysis buffer (50 mM Tris pH 8, 150 mM K-acetate, 2 mM

Mg-acetate, 1 mM ethylene glycol-bis(β-aminoethyl ether)-N,N,N',N'-tetraacetic acid (EGTA), 10% glycerol) with protease inhibitor cocktail (Roche), 1 mM DTT, 1 mM PMSF, and DNAseI. Cells were then passed through an Emulsiflex press and cleared by centrifugation at 23,000 × g for 20 min. For baculovirus expression of sfGFP-MAP7 or TagRFP-MAP7, the Bac-to-Bac protocol (Invitrogen) was followed. SF9 cells were grown in shaker flasks to ~2 × 10[6]/mL and infected at a ratio of 10 mL virus to 250 mL cells. The infection was allowed to proceed for 48 hr before cells were harvested and frozen in LN₂. Cell pellets were resuspended in lysis buffer (50 mM Tris-HCl, pH 8.0, 150 mM K-acetate, 2 mM Mg-acetate, 1 mM EDTA, 10% glycerol, 0.1 mM ATP) with protease inhibitor cocktail (Roche). Cells were lysed by addition of 1% Triton X-100 for 10 min on ice. Clarified lysate from either bacterial or baculovirus expression was passed over a column with Strep-tactin Superflow resin (Qiagen). After incubation, the column was washed with four column volumes of lysis buffer, then bound proteins were eluted with 3 mM desthiobiotin (Sigma) in lysis buffer. Eluted proteins were concentrated on Amicon concentrators and passed through a superose-6 or superdex-200 (GE Healthcare) gel-filtration column in lysis buffer using a Bio-Rad NGC system. Peak fractions were collected, concentrated, and flash frozen in LN₂. Protein concentration was determined by measuring the absorbance of the fluorescent protein tag and calculated using the molar extinction coefficient of the tag. Purified BicD2N was used to isolate DDB complexes from rat brain cytosol as previously described[37]. DDB complexes were labeled with 5 μM SNAP-TMR dye during the isolation procedure, and were frozen in small aliquots and stored at −80 °C. The resulting preparations were analyzed by SDS polyacrylamide gel electrophoresis (SDS-PAGE).

**Co-sedimentation assays**. For co-sedimentation assays, microtubules were prepared by polymerizing 25 mg/mL of porcine tubulin in assembly buffer (BRB80 buffer supplemented with 1 mM GTP, 1 mM DTT) at 37 °C for 15 min, then a final concentration of 20 μM taxol was added to the solution, which was incubated at 37 °C for an additional 15 min. Microtubules were pelleted over a 25 % sucrose cushion at 100,000 g at 25 °C for 10 min, then resuspended resuspended in BRB80 buffer with 1 mM DTT and 10 μM taxol. Binding reactions were performed by mixing 500 nM of sfGFP-MAP7 or sfGFP-tau (that had been pre-centrifuged at 100,000 g) with the indicated concentrations of microtubules in assay buffer (50 mM Tris pH 8, 150 mM K-acetate, 2 mM Mg-acetate, 1 mM EGTA, 10% glycerol and supplemented with 1 mM DTT, 10 μM taxol, and 0.01 mg/mL BSA) and incubated at 25 °C for 20 min. The mixtures were then pelleted at 90,000 × g at 25 °C for 10 min. Supernatant and pellet fractions were recovered, resuspended in sample buffer, and analyzed by SDS-PAGE. Protein band intensities were quantified using ImageJ. The uncropped images of the SDS-PAGE gels are shown in Supplementary Figure 5.

**Pull-down assays**. Pull-down assays were performed with either sfGFP-MAP7 or sfGFP-ensconsin tagged at the C-terminal end with a DYKDDDDK (FLAG) epitope. FLAG beads (Thermofisher) were washed into assay buffer (50 mM Tris, pH 8, 150 mM K-acetate, 2 mM Mg-acetate, 1 mM EGTA, 10% glycerol and supplemented with 1 mM DTT and 0.1 mg/mL bovine serum albumin), then incubated with 500 nM MAP7, 500 nM ensconsin, or buffer (beads alone control) for 1 h rotating at 4 °C. FLAG beads were then washed in assay buffer five times, then resuspended in assay buffer and 500 nM (for ensconsin) or 1 μM K560 (for MAP7) was added to the beads alone control and the experimental condition. The 350 μL final volume solutions were incubated for 1 h rotating at 4 °C. The supernatants were collected, then the bead pellets were washed five times in assay buffer and resuspended in one bead bed volume. Pull-down assays were performed with sfGFP-tau using GFP-binding protein (GBP) beads. GBP beads were washed into assay buffer (60 mM Hepes pH 7.4, 150 mM K-acetate, 2 mM Mg-acetate, 1 mM EGTA, and 10% glycerol) supplemented with, 0.5% Pluronic F-168, and 0.2 mg/mL κ-casein, 10 μM taxol, 1 mM DTT and 0.1 mg/mL BSA), then incubated with 500 nM sfGFP-tau or buffer (beads alone control) for 1 h rotating at 4 °C. GBP beads were then washed in assay buffer five times, then resuspended in assay buffer and 250 nM RFP-MAP7 was added to the beads alone control and the experimental condition. The 350 μL final volume solutions were incubated for 1 h rotating at 4 °C. The supernatants were collected, then the bead pellets were washed five times in assay buffer and resuspended in one bead bed volume. Gel samples of the supernatants and pellets were analyzed by SDS-PAGE. The supernatant samples that were run on SDS-PAGE were 15% of the pellet samples.

**TIRF microscopy**. For TIRF-M experiments, a mixture of native tubulin, biotin-tubulin, and fluorescent-tubulin purified from porcine brain (~10:1:1 ratio) was assembled in BRB80 buffer (80 mM PIPES, 1 mM MgCl₂, 1 mM EGTA, pH 6.8 with KOH) with 1 mM GTP for 15 min at 37 °C, then polymerized MTs were stabilized with 20 μM taxol. Microtubules were pelleted over a 25% sucrose cushion in BRB80 buffer to remove unpolymerized tubulin. Flow chambers containing immobilized microtubules were assembled as described[37]. Imaging was performed on a Nikon Eclipse TE200-E microscope equipped with an Andor iXon EM CCD camera, a × 100, 1.49 NA objective, four laser lines (405, 491, 568, and 647 nm) and Micro-Manager software[43]. All MAP7 and tau competition experiments were

performed in assay buffer (30 mM Hepes pH 7.4, 150 mM K-acetate, 2 mM Mg-acetate, 1 mM EGTA, and 10% glycerol) supplemented with 0.1 mg/mL biotin-BSA, 0.5% Pluronic F-168, and 0.2 mg/mL κ-casein (Sigma).

For all competition experiments, one protein was held at a constant concentration (50 nM sfGFP- or TagRFP-MAP7 or 100 nM mTagBFP- or sfGFP-tau), whereas the other was increased; both proteins were premixed and flowed into the chamber at the same time. Competition experiments were performed with a mixture of either sfGFP-MAP7 and mTagBFP-tau or a mixture of TagRFP-MAP7 and sfGFP-tau. For live imaging, images were taken every 15 seconds for a total of 10 min. For fluorescence intensity analysis, ImageJ was used to draw a line across the microtubule of either the Tau or MAP7 channel and the integrated density was measured. The line was then moved adjacent to the microtubule of interest and the local background was recorded. The background value was then subtracted from the value of interest to give a corrected intensity measurement. For dwell times, either MAP7 or tau was diluted to single-molecule levels in the above-mentioned assay buffer and imaged at 0.5 s/frame and 0.128 sec/frame, respectively. The dwell times were measured by kymograph analysis in ImageJ by drawing a rectangle box around the beginning and end of a single dwelling molecule. The cumulative frequencies of MAP7 and tau dwell times were fit to a one phase exponential decay to derive the decay constants ($\tau$) for each MAP.

All MAP7-kinesin construct motility and recruitment assays were performed in BRB80 buffer (80 mM PIPES pH 6.8, 1 mM MgCl₂ and 1 mM EGTA) supplemented with 1 mM ATP, 150 mM K-acetate, with 0.1 mg/mL biotin-BSA, 0.5% Pluronic F-168, and 0.2 mg/mL κ-casein. MAP7 was flowed in first, followed by K560, K523, K508, and K490. Fluorescence intensity analysis was performed as described above. Kymographs were made from movies of K560 in the absence and presence of MAP7 and velocity and processivity parameters were measured for individual K560 runs that moved processively for ≥ 250 nm at a speed of ≥ 80 nm/sec. Velocity data were fit with a Gaussian equation and the cumulative frequencies of K560 run lengths were fit to a one phase exponential decay to derive the mean decay constant ($\tau$).

MAP7-tau-K560 experiments were carried out in assay buffer (30 mM Hepes pH 7.4, 150 mM K-acetate, 2 mM Mg-acetate, 1 mM EGTA, and 10% glycerol) supplemented with 1 mM ATP, 0.1 mg/mL biotin-BSA, 0.5% Pluronic F-168, and 0.2 mg/mL κ casein. MAP7 and tau were flowed in simultaneously, followed by K560. Kymographs were analyzed at regions of microtubules where there were clear tau patches that K560 molecules encountered. Events of pausing, passing, or detaching from the tau patch were counted. MAP7-KIF1A, tau-KIF1A, and MAP7-DDB experiments were performed in assay buffer (30 mM Hepes pH 7.4, 150 mM K-acetate, 2 mM Mg-acetate, 1 mM EGTA, and 10% glycerol) supplemented with 1 mM ATP, 0.1 mg/mL biotin-BSA, 0.5% Pluronic F-168, and 0.2 mg/mL κ-casein. Fluorescence intensity analysis and velocity measurements were performed as described above. Events of pausing, passing, or detaching from the tau patch were counted for KIF1A.

**Fly stocks**. We used the Gal4 driver line, 221-Gal4[29], to drive expression of UAS-CD4-RFP[29] to visualize the dendrite morphology of class I DA neurons, UAS-manII-eGFP[29] to visualize Golgi outposts, UAS-ensconsin-mCardinal (this study) and UAS-GFP-tau[44]. We generated the UAS-ensconsin-mCardinal fly line by cloning the full-length ensconsin gene from the DGRC (#1070880) into a pUASt vector with an mCardinal tag on the C-terminus. Injection services were performed by Rainbow Transgenics. The endogenous Mi[MIC] GFP-tau line (Stock 60199) was obtained from the Bloomington Drosophila Stock Center (Department of Biology, Indiana University, Bloomington, IN).

**Live imaging and analysis**. Whole, live third instar larvae were mounted in 90% glycerol under coverslips sealed with grease, and imaged using an Olympus FV1000 laser scanning confocal microscope. For morphological and Golgi outpost distribution analysis, the following genotypes were imaged: (1) 221-Gal4, UAS-CD4-RFP, (2) 221-Gal4, UAS-CD4-RFP; UAS-GFP-tau, (3) 221-Gal4, UAS-CD4-RFP; UAS-ens-mCardinal, (4) 221-Gal4, UAS-CD4-RFP; UAS-GFP-tau/UAS-ens-mCardinal, (5) 221-Gal4, UAS-CD4-RFP;UAS-manII-eGFP, and (6) 221-Gal4, UAS-CD4-RFP;UAS-manII-eGFP/UAS-ens-mCardinal. Z-stacks containing the dendritic arbors of the class I DA sensory neurons were collected for analysis from segment A3 or A4. Total dendrite length and number of branchpoints were determined from maximum Z-projections of the Z-stack image files using the Simple Neurite Tracer plugin[45] for ImageJ Fiji.

**Immunohistochemistry and immunocytochemistry**. For immunohistochemical staining on third larval instar fillets, third instar larvae were dissected and filleted in pre-chilled PBS, followed by fixation in 4% paraformaldehyde for 20 min at room temperature. The fillets were permeabilized in phosphate-buffered saline (PBS) with 0.3% Triton X-100 (PBS-TX), blocked with 5% BSA in PBS for 1 h at room temperature, then incubated overnight at 4 °C with primary antibodies. The primary antibodies used were chicken anti-GFP (1:500; GFP-1020, Aves Labs, RRID: AB_10000240), goat anti-HRP-Cy5 (1:1000; 123-605-021, Jackson ImmunoResearch, RRID:AB_2338967) and rabbit anti-ensconsin 172 (1:500; gift from R. Giet). For the ensconsin antibody staining, it was essential that larval muscle be cleared entirely from the fillet in order to get staining of the neurons, because this

antibody prominently stains the muscles. Secondary antibodies consisted of appropriate fluorescence-conjugated anti-donkey IgG (1:1000; Jackson ImmunoResearch). Fillets were incubated with the secondary antibodies for 2 hrs at room temperature before being mounted using VectaShield mounting medium (Vector Laboratories). Slides were imaged on an Olympus FV1000 laser scanning confocal microscope using an oil immersion ×40 or ×60 objective.

Neuronal cultures were isolated from mouse cortices[46] and cultured for 4 days in vitro. Neurons were dissociated from mouse cortices, plated in Dulbecco's

Modified Eagle Medium with 10% fetal bovine serum and 1mM L-glutamine, and grown at 37 °C with 5% $CO_2$ for 4 days before fixation. The cultures were fixed in 4% paraformaldehyde for 20 min at room temperature, washed several times with PBS, permeabilized in PBS with 0.3% Triton X-100 (PBS-TX), and blocked with 5% BSA in PBS for 1 h at room temperature. Cultures were then incubated overnight at 4 °C with primary antibodies at a concentration of 1:300 for rabbit anti-MAP7 (Thermofisher PA5-31782), 1:500 for mouse monoclonal anti-Tau (Millipore MAB3420), 1:1000 for mouse anti-alpha Tubulin (Sigma Clone DM1A T9026), or

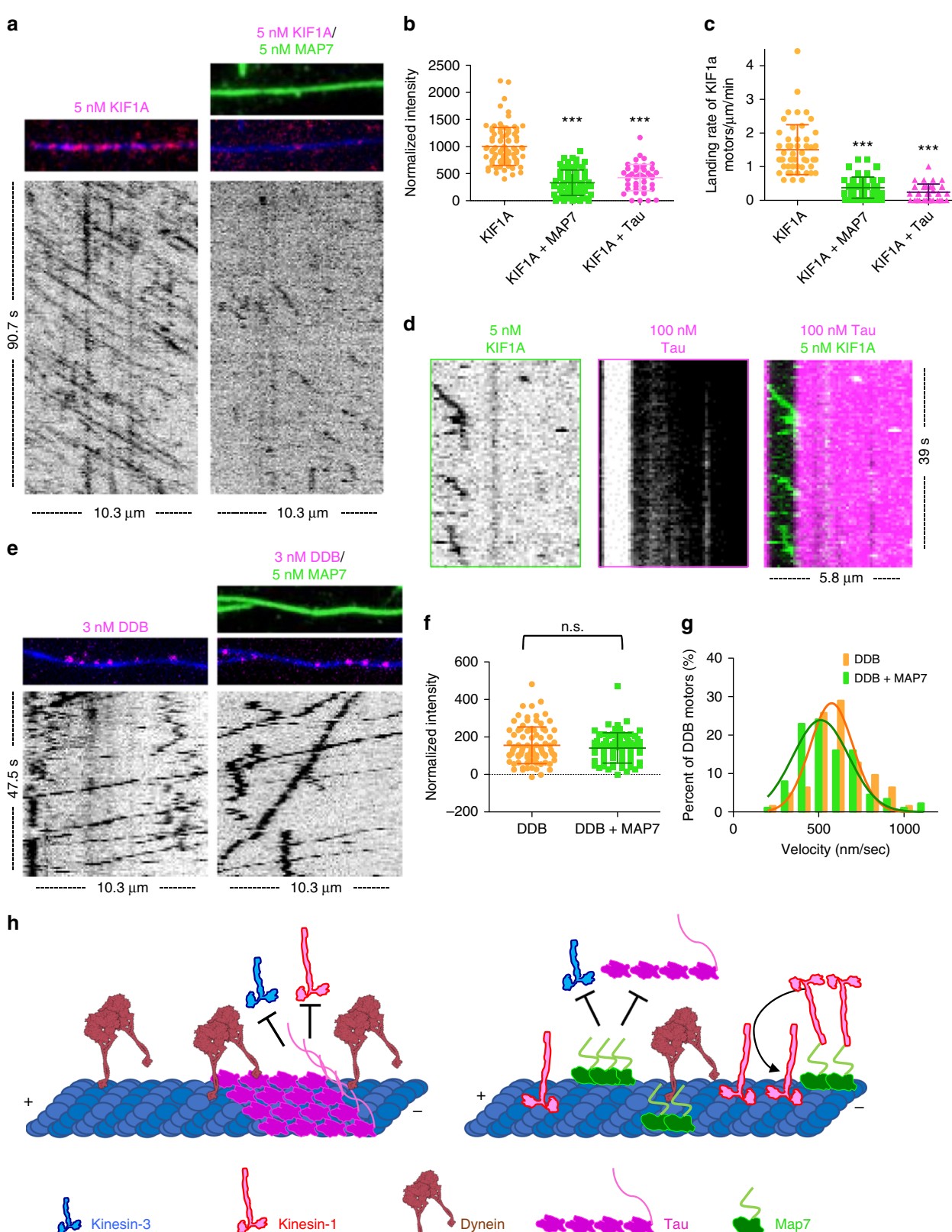

**Fig. 6** MAP7 and tau differentially affect other microtubule motors. **a** TIRF-M images and corresponding kymographs of 5 nM KIF1A-mScarlet (kinesin-3) in the absence and presence of 5 nM sfGFP-MAP7. Images are 10.3 μm in length. **b** Quantification of KIF1A fluorescence intensity in the absence and presence of 5 nM sfGFP-MAP7 or 100 nM mTagBFP-tau (means ± s.d. are 1003.1 ± 351.1, 332.2 ± 235.2, and 427.1 ± 243.7 A.U., and $n = 89$, 69, and 46 microtubules for KIF1A alone, KIF1A + MAP7, and KIF1A + tau, respectively, from three independent trials). **c** Quantification of KIF1A landing rates in the absence and presence of sfGFP-MAP7 or mTagBFP-tau (means ± s.d. are 1.50 ± 0.74, 0.38 ± 0.31, and 0.24 ± 0.24 motors μm$^{-1}$ min$^{-1}$, and $n = 50$, 51, and 38 for KIF1A alone, KIF1A + MAP7, and KIF1A + tau, respectively, from three independent trials). Both graphs are scatterplots with all datapoints plotted and lines representing the means ± s.d. **d** Kymograph depicting KIF1A-mScarlet motors encountering an mTagBFP-tau patch. **e** TIRF-M images and corresponding kymographs of 3 nM dynein–dynactin-BicD2 (DDB)-TMR + 1 mM ATP in the absence and presence of 5 nM sfGFP-MAP7. **f** Quantification of DDB-TMR fluorescence intensity in the absence and presence of sfGFP-MAP7 (means ± s.d. are 164.3 ± 103.7 and 147.3 ± 85.7 A.U. and $n = 89$ and 70 microtubules for DDB alone and DDB + MAP7, respectively from two independent trials; $P = 0.27$). **g** Velocity histograms of DDB-TMR + 1 mM ATP in the absence and presence of sfGFP-MAP7 with Gaussian fits. Means ± s.d. are 606.4 ± 178.0 and 555.2 ± 186.5 nm/sec and $n = 62$ and 87 motors for DDB alone and DDB + MAP7, respectively from three independent trials ($P = 0.095$). All experiments were repeated with at least two separate protein preparations for each protein. A student's $t$-test was used for all statistical analyses. *** indicates $P < 0.0001$. **h** Model for how competition between MAP7 and tau directs motor transport. In the presence of tau (pink), kinesin-1 (red), and kinesin-3 (blue) are inhibited from the microtubule, whereas dynein (burgundy) motility is unperturbed. The presence of MAP7 (green) evicts tau from the microtubule lattice, facilitating kinesin-1 recruitment without altering dynein activity. MAP7 inhibits kinesin-3 similarly to tau, ensuring the exclusion of kinesin-3 from microtubules in tau- and MAP7-rich environments

1:1000 for rabbit anti-beta Tubulin (Abcam ab6046). Secondary antibodies were used at 1:1000 for Cy3 donkey anti-rabbit or Cy5 donkey anti-mouse and incubated for 1 h at room temperature. Cells were then rinsed several times with PBS and mounted using VectaShield mounting medium (Vector Laboratories).

**ATPase assays**. ATPase assays were performed using an adenosine triphosphate/nicotinamide adenine dinucleotide (ATP/NADH) coupled method in assay buffer containing 80 mM PIPES, pH 6.8, 1 mM MgCl2, and 1 mM EGTA, supplemented with 0.1% Triton X, 1 mM DTT, and 0.1 mg/mL BSA. The A340 was measured at 37 °C for 5 min in the presence of assay buffer with 50 nM K560, 2 mM ATP, 0.2 mM NADH (Roche), 2 mM PEP (in 8 mM KOH), 0.02% pyruvate kinase/lactate dehydrogenase (enzymes from rabbit muscle, Sigma), and at a range of taxol-stabilized microtubule concentrations. Both mTagBFP-tau and sfGFP-tau were used in this assay, as well as sfGFP-MAP7 that was purified from insect cells using the baculovirus protocol. For the assays containing MAP7 and/or tau with K560, MAP7 and/or tau was incubated with microtubules for 5 min before K560 was added and the absorbance was measured. ATPase rate was measured and divided by the K560 (ATPase) concentration to determine the molar activity of K560 per second. The ATPase rates were corrected for background NADH decomposition of controls containing no K560. MAP7 and tau did not exhibit ATPase activity on their own either in the presence of absence of microtubules. The resulting molar activity per second was plotted at a range of microtubule concentrations, and Michaelis–Menten curves were fit to the data to derive $K_m$ and $V_{max}$ values.

**Statistical analysis**. All statistical tests were performed with a two-tailed unpaired Student's $t$-test.

**Data availability**. The data that support the findings of this study are available from the corresponding author upon reasonable request.

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

## Acknowledgements

We thank Ruensern Tan and Richard McKenney for providing the purified DDB complex, and Richard McKenney for useful discussion and critical reading of the manuscript. We thank Regis Giet for generously providing us with the ensconsin antibody and for useful discussion. We thank Scott Cameron and Kim McAllister for providing the fixed mouse neuronal cultures. We also thank the DGRC for providing the *ens* cDNA construct, and the Bloomington Drosophila Stock Center for providing fly stocks. This work was supported by the March of Dimes Basil O'Connor Award and NIH grant 1R00HD080981 to K.M.O.M. This material is based upon work supported by the National Science Foundation Graduate Research Fellowship Program under Grant No. 1650042 to B.Y.M. Any opinions, findings, and conclusions or recommendations expressed in this material are those of the authors and do not necessarily reflect the views of the National Science Foundation.

## Author contributions

B.Y.M. and K.M.O.M. designed the experiments and wrote the manuscript. B.Y.M., D.S., and T.T. purified the recombinant proteins. B.Y.M. performed the in vitro TIRF-M experiments. B.Y.M. and K.M.O.M. performed the neuronal culture, microtubule co-pelleting, and MAP7/Ens pull-down experiments and analyzed data. M.B. and K.M.O.M. performed the *Drosophila* work and analyzed data. D.S. and B.A. performed the ATPase assays and analyzed data.
