## [peer review file · Nature Communications]

Reviewer #1 (Remarks to the Author):

This interesting paper examines the roles of structural microtubule (MT)-associated proteins (MAPs) in regulation of MT-based transport. The authors investigate how binding of two common MAPs, tau and MAP7, to MT lattice influence motility of MT motor proteins *in vitro* and *in vivo*. These MAPs have been selected because of the known differences in their effects on motility of conventional MT motor protein kinesin-1. The authors show that MAP7 and tau compete for binding to MTs *in vitro* and that MAP7 can displace tau from MT lattice because of its higher MT binding activity, and longer MT residence time. The negatively charged C-terminal domain of MAP7 plays important role in the tau displacement. Overexpression tau and MAP7 homologue *ensconsin* in *Drosophila* larval dendritic arborization neurons have opposite effects on the numbers of dendritic branches or distribution of MT motor cargoes. These effects are consistent with inhibition of kinesin-1 motility by tau and stimulation by MAP7 in single molecule assays *in vitro*. MAP7 directly binds kinesin-1 end enhances its motility by facilitating interaction of this motor protein with MTs. The authors also show that binding to MTs of tau or MAP7 inhibits motility of another member of kinesin family kinesin-3 but has no effect on dynein/dynactin motility. Overall, this is important study that shows how competition among structural MAPs for binding to MTs can selectively affect motility of individual MT motor proteins and therefore MT-based transport in cells. The paper is clearly written, and experimental data are of high quality. This work will be likely of significant interest to broad scientific readership of *Nature Communications*. However, the paper can not be published as written and presented.

The authors indicate that full-size MAP7 removes tau from MTs but truncated MAP7 Δ C that lacks conserved negatively charged C-terminal region binds to MTs but does not remove tau. However, Fig. 2h shows that truncated MAP7 apparently displaced tau as well but at a significantly reduced rate (Fig. 2h). The paper would benefit from further analysis of the mechanism underlying competition between the two MAPs. Whether C-terminal region of MAP7, similar to MAP7D3, has a separate MT binding site that enhances binding to MTs (a possibility that should be tested by determining KD for MAP7 Δ C) or direct interaction of MAP7 C-terminus with tau is responsible for the tau displacement?

The authors use truncated version of kinesin-1 K560 to study interaction with *ensconsin* and MAP7 and show that amino acid residues 500-523 of kinesin-1 are involved in this interaction. K560 construct terminates in the hinge-2 region and therefore lacks C-terminal tail domain involved in autoinhibition. However, a recent study (Barlan et al., 2013) indicates that kinesin-1 mutants without autoinhibitory function do not require *ensconsin* for transport and suggests that *ensconsin* activates kinesin-1 by relieving autoinhibition. Furthermore, pull-down experiments performed in this study failed to detect interaction between *ensconsin* and kinesin-1 *in vitro*. The authors should address discrepancy between their data and the results of Barlan et al. (2013).

Minor comments:

Panels b and c in legend to Supplemental Fig 2 are mislabeled.

Top right panel in fig. 4k seems to be mislabeled.

Typographical and grammatical errors should be corrected.

Reviewer #2 (Remarks to the Author):

The paper by Monroy and colleagues describes competitive interactions between two well-known microtubule-binding proteins, tau and MAP7/ensconsin. The authors beautifully demonstrated that MAP7 promotes kinesin-1 interaction with microtubules, significantly stimulating microtubule-dependent ATPase activity of kinesin. Analysis of this effect using TIRF microscopy shows that the landing rate of kinesin on microtubules coated with MAP7 is significantly higher than on microtubules that do not contain it. Two-color TIRF data and co-sedimentation analysis show MAP7 has a higher affinity for microtubules than tau and displaces microtubule-bound tau, allowing kinesin to bind microtubules and initiate movement. Based on these beautiful in vitro data, the authors suggest a model that explains how kinesin is activated in axons containing tau and how inhibition of kinesin-3 by MAP7 explains its exclusion from the axons. I have no problem with in vitro results; these studies are well done and the data are novel and of a very high quality.

Unfortunately, the in vivo experiments do not quite match the quality of the biochemical work. First, static images of Golgi outpost distribution in DA neurons are used in the paper as a proxy for cargo transport. I can see several problems with this approach. First, it is not quite clear how outposts assume their steady-state distribution. It is possible that they are excluded from dendrites by kinesin-1, but the situation is complicated by the fact that not all microtubules are uniformly polarized in dendrites. Furthermore, as the authors know well, Golgi outposts themselves can nucleate microtubules which complicates interpretation of the results. Moreover, even if indeed ensconsin overexpression activates kinesin-1 in vivo, as the authors suggested, it is not a new piece of information, as ensconsin requirement for kinesin-1 activity in vivo has already been demonstrated in at least two publications. What makes this paper exciting is not MAP7-dependent kinesin requirement but a new concept of MAP7-tau competition that is so beautifully developed in the paper.

I want to emphasize once again, that biochemical data in the paper are novel and very interesting. These data alone are sufficient for a good publication, so my suggestion would be to publish biochemical results without the in vivo results presented in Fig. 3.

This is a nice story that should be published soon.

Reviewer #3 (Remarks to the Author):

This manuscript addresses the relationship between MAP7 and tau activities on microtubules and consequences on the balance and distribution of microtubule motor proteins. Using biochemistry approaches and TIRF microscopy, the authors find that MAP7 and tau compete for microtubule binding in vitro and propose a mechanism by which MAP7 displaces tau from the lattice. They further show that MAP7 stimulates kinesin-based transport in vivo and enhances the interaction of kinesin-1 with microtubules in vitro, at the difference of the known inhibitory effect of tau on kinesin-1 motility. They also demonstrate that both MAP7 and tau inhibit the dendritic-specific motor kinesin-3 and that MAP7 has no effect on dynein in vitro. Based on their data, the authors propose a model explaining how the interplay between MAP7 and tau on the microtubule surface determines the distribution and activity of motor proteins. Overall I find this paper very interesting with data of high quality. It provides novel mechanistic insights into the regulation of motors by the combined activities of MAPs, a process that remains poorly understood.

However, there are some questions/clarifications that should be addressed prior to publication:

1/ Regarding the competition between MAP7 and tau, a truncated version of MAP7 containing the MTBD (MAP7 Δ C) is not able to displace tau from the microtubule surface (Fig. 2h). In that case, both MAP7 Δ C and tau can bind along the microtubule wall (Fig. 2g), which is not in favour of a same microtubule-binding site for the two proteins. How do the author explain this?

2/ Based on their results obtained with the mutant MAP7 Δ C, the authors propose that the C-terminal part of MAP7 plays a role in displacing tau molecules from microtubules. The authors should confirm this hypothesis by evaluating the specific effect of the C-terminal part of MAP7 on tau interaction along microtubules.

3/ Did the author check that MAP7 and tau interact together? Indeed, the formation of a complex between MAP7 and tau (for instance via the C-terminal part of MAP7) could also participate to the inhibition of tau binding to microtubules in the presence of MAP7.

4/ About the apparent K_d of MAP7 and tau for microtubules, why didn't the authors use the same tubulin concentrations for the two proteins (a smaller number of concentrations has been used to determine the binding curve of tau, Fig. 2j)?

The apparent K_d of tau for microtubules seems higher compared to the one previously determined in the literature by similar assays holding tau constant and varying tubulin concentrations (about 0.1-0.3 μM , see for instance Duan and Goodson, 2012, *Mol Biol Cell*, 23: 4796-806; Elie et al, 2015, *Sci Rep*, 5:9964). The fact that tau binds microtubules before MAP7 (Fig. 2e) also suggests a higher affinity of tau for microtubules compared to MAP7, and thus a lower K_D value.

How do the authors explain these differences?

5/ The authors reconstitute the behaviour of kinesin-1 along microtubules in the presence of both MAP7 and tau (Supplementary Fig. 3 and Fig. 4k). They show that MAP7 (50 nM) does not change the behaviour of K560 (10 nM) when encountering a patch of tau (100 nM) bound to microtubules, although MAP7 is still able in these conditions to recruit K560 outside the tau patches (Fig. 4k). What happens if the authors increase the amount of MAP7? By varying MAP7 concentrations, tau should progressively detach from the microtubules, restoring K560 processivity along microtubules. Such an experiment would nicely recapitulate part of the model proposed by the authors in Fig. 5h.

Minor points:

- Regarding the single molecule analysis (Fig. 2k), it is very difficult to see on the graph the cumulative frequency of tau dwell times. The author should present the data obtained for tau and MAP7 on two separate graphs with an appropriate scale for the dwell-time axis.
- The significance of differences between some data sets is lacking on several graphs (e.g. Fig. 4 h and j, Fig. 5b and c)

Point-by-Point Responses to Reviewers

Reviewer #1 (Remarks to the Author):

This interesting paper examines the roles of structural microtubule (MT)-associated proteins (MAPs) in regulation of MT-based transport. The authors investigate how binding of two common MAPs, tau and MAP7, to MT lattice influence motility of MT motor proteins in vitro and in vivo. These MAPs have been selected because of the known differences in their effects on motility of conventional MT motor protein kinesin-1. The authors show that MAP7 and tau compete for binding to MTs in vitro and that MAP7 can displace tau from MT lattice because of its higher MT binding activity, and longer MT residence time. The negatively charged C-terminal domain of MAP7 plays important role in the tau displacement. Overexpression tau and MAP7 homologue ensconsin in *Drosophila* larval dendritic

arborization neurons have opposite effects on the numbers of dendritic branches or distribution of MT motor cargoes. These effects are consistent with inhibition of kinesin-1 motility by tau and stimulation by MAP7 in single molecule assays in vitro. MAP7 directly binds kinesin-1 end enhances its motility by facilitating interaction of this motor protein with MTs. The authors also show that binding to MTs of tau or MAP7 inhibits motility of another member of kinesin family kinesin-3 but has no effect on dynein/dynactin motility. Overall, this is important study that shows how competition among structural MAPs for binding to MTs can selectively affect motility of individual MT motor proteins and therefore MT-based transport in cells. The paper is clearly written, and experimental data are of high quality. This work will be likely of significant interest to broad scientific readership of Nature Communications. However, the paper can not be published as written and presented.

We thank the reviewer for their supportive comments about our work.

The authors indicate that full-size MAP7 removes tau from MTs but truncated MAP7 Δ C that lacks conserved negatively charged C-terminal region binds to MTs but does not remove tau. However, Fig. 2h shows that truncated MAP7 apparently displaced tau as well but at a significantly reduced rate (Fig. 2h).

After 10 minutes of imaging, tau intensity on the microtubule is reduced by 70% in the presence of full-length MAP7 compared to a reduction of 27% in the presence of MAP7 Δ C. We have now revised the manuscript for accuracy to state that removing the C-terminus of MAP7 greatly reduces its ability to evict tau from the microtubule (page 5).

The paper would benefit from further analysis of the mechanism underlying competition between the two MAPs. Whether C-terminal region of MAP7, similar to MAP7D3, has a separate MT binding site that enhances binding to MTs (a possibility that should be tested by determining KD for MAP7 Δ C) or direct interaction of MAP7 C-terminus with tau is responsible for the tau displacement?

We agree with the reviewer and have performed the suggested experiments to strengthen the mechanism of competition. We now show that the C-terminus of MAP7 does not bind microtubules on its own (new Supplemental Figure 2d-e). In fact, removing the C-terminus of MAP7 slightly increases the apparent binding affinity of MAP7 for microtubules (new Supplemental Figure 2b). The C-terminal half of MAP7 is highly negatively charged and may be repelled by the negatively charged C-terminal tails of tubulin. We also performed a pull-down between purified MAP7 and tau and have found that they do not interact with one another in solution (new Supplemental Figure 2c), suggesting this is not a mechanism for tau eviction. We thank the reviewer for suggesting these experiments and strengthening the paper.

The authors use truncated version of kinesin-1 K560 to study interaction with ensconsin and MAP7 and show that amino acid residues 500-523 of kinesin-1 are involved in this interaction. K560 construct terminates in the hinge-2 region and therefore lacks C-terminal tail domain involved in autoinhibition. However, a recent study (Barlan et al., 2013) indicates that kinesin-1 mutants without autoinhibitory function do not require ensconsin for transport and suggests that ensconsin activates kinesin-1 by relieving autoinhibition. Furthermore, pull-down experiments performed in this study failed to detect interaction between ensconsin and kinesin-1 in vitro. The authors should address discrepancy between their data and the results of Barlan et al. (2013).

Our pull-downs with purified proteins only suggest a modest interaction between MAP7/ensconsin with K560. We use the word “modest” in the text (page 7), because the interaction between MAP7 and K560

appears relatively weak in solution, these data may indicate a more transient interaction rather than a stable association of these two proteins. This could be why the pull-downs from lysates in Barlan et al. (2013) did not detect a robust interaction. Combined with our other data that indicate MAP7 does not tether K560 to the microtubule (Fig. 4f), our model is consistent with and expands upon the proposed model of Barlan et al. (2013), which we highlight on page 9.

It is possible that MAP7/ens could recruit as well as relieve the autoinhibition of full-length kinesin-1, and we have now added additional text in the discussion to address this point. It is very clear that MAP7/ens recruits non-auto-inhibited K560 to the microtubule, but it would be interesting in the future to test if MAP7/ens directly relieves the autoinhibition of kinesin-1.

Minor comments:

Panels b and c in legend to Supplemental Fig 2 are mislabeled.

Top right panel in fig. 4k seems to be mislabeled.

Typographical and grammatical errors should be corrected.

We thank the reviewer for catching these mistakes. We have rearranged a few figures and have thoroughly gone through the figures and manuscript to ensure proper labeling of figures and correction of grammatical errors.

Reviewer #2 (Remarks to the Author):

The paper by Monroy and colleagues describes competitive interactions between two well-known microtubule-binding proteins, tau and MAP7/ensconsin. The authors beautifully demonstrated that MAP7 promotes kinesin-1 interaction with microtubules, significantly stimulating microtubule-dependent ATPase activity of kinesin. Analysis of this effect using TIRF microscopy shows that the landing rate of kinesin on microtubules coated with MAP7 is significantly higher than on microtubules that do not contain it. Two-color TIRF data and co-sedimentation analysis show MAP7 has a higher affinity for microtubules than tau and displaces microtubule-bound tau, allowing kinesin to bind microtubules and initiate movement. Based on these beautiful in vitro data, the authors suggest a model that explains how kinesin is activated in axons containing tau and how inhibition of kinesin-3 by MAP7 explains its exclusion from the axons. I have no problem with in vitro results; these studies are well done and the data are novel and of a very high quality.

Unfortunately, the in vivo experiments do not quite match the quality of the biochemical work. First, static images of Golgi outpost distribution in DA neurons are used in the paper as a proxy for cargo transport. I can see several problems with this approach. First, it is not quite clear how outposts assume their steady-state distribution. It is possible that they are excluded from dendrites by kinesin-1, but the situation is complicated by the fact that not all microtubules are uniformly polarized in dendrites. Furthermore, as the authors know well, Golgi outposts themselves can nucleate microtubules which complicates interpretation of the results. Moreover, even if indeed ensconsin overexpression activates kinesin-1 in vivo, as the authors suggested, it is not a new piece of information, as ensconsin requirement for kinesin-1 activity in vivo has already been demonstrated in at least two publications. What makes this paper exciting is not MAP7-dependent kinesin requirement but a new concept of MAP7-tau competition that is so beautifully developed in the paper.

I want to emphasize once again, that biochemical data in the paper are novel and very interesting. These data alone are sufficient for a good publication, so my suggestion would be to publish biochemical results without the in vivo results presented in Fig. 3.

This is a nice story that should be published soon.

We thank the reviewer for their thorough and honest evaluation of the manuscript. We feel the inclusion of the *in vivo* data, while somewhat correlative and redundant with prior work, is still important for the overall message of the paper and supportive of the *in vitro* work presented. The microtubule orientation in the DA neurons is uniform in the primary branches (plus ends growing towards the cell body) and uniform in the terminal branches (plus ends growing towards the distal dendrite tips) (Stone et al., 2008; Ori-McKenney et al., 2012), thus the final placement of the Golgi outposts upon *ensconsin* overexpression is consistent with kinesin driving these cargo towards the location of plus ends. Even upon Golgi outpost nucleation of microtubules from the distal tips, for example, the enhancement of kinesin-1 activity by *ensconsin* overexpression would drive Golgi outposts and other cargo back towards the uniform tracks in the primary branches and eventually to the cell body, where the plus-ends are localized. We have modified our text to clarify a few of these points in support of the *in vivo* data.

Reviewer #3 (Remarks to the Author):

This manuscript addresses the relationship between MAP7 and tau activities on microtubules and consequences on the balance and distribution of microtubule motor proteins. Using biochemistry approaches and TIRF microscopy, the authors find that MAP7 and tau compete for microtubule binding *in vitro* and propose a mechanism by which MAP7 displaces tau from the lattice. They further show that MAP7 stimulates kinesin-based transport *in vivo* and enhances the interaction of kinesin-1 with microtubules *in vitro*, at the difference of the known inhibitory effect of tau on kinesin-1 motility. They also demonstrate that both MAP7 and tau inhibit the dendritic-specific motor kinesin-3 and that MAP7 has no effect on dynein *in vitro*. Based on their data, the authors propose a model explaining how the interplay between MAP7 and tau on the microtubule surface determines the distribution and activity of motor proteins. Overall I find this paper very interesting with data of high quality. It provides novel mechanistic insights into the regulation of motors by the combined activities of MAPs, a process that remains poorly understood.

However, they are some questions/clarifications that should be addressed prior to publication:

1/ Regarding the competition between MAP7 and tau, a truncated version of MAP7 containing the MTBD (MAP7 Δ C) is not able to displace tau from the microtubule surface (Fig. 2h). In that case, both MAP7 Δ C and tau can bind along the microtubule wall (Fig. 2g), which is not in favour of a same microtubule-binding site for the two proteins. How do the author explain this?

Although MAP7 Δ C does not displace tau to the same extent as full-length MAP7, the binding of MAP7 Δ C and tau is still mutually exclusive on the lattice, indicating they are still competing for overlapping binding sites on the microtubule. We have clarified this point in the text.

2/ Based on their results obtained with the mutant MAP7 Δ C, the authors propose that the C-terminal part of MAP7 plays a role in displacing tau molecules from microtubules. The authors should confirm this hypothesis by evaluating the specific effect of the C-terminal part of MAP7 on tau interaction along microtubules.

We have performed additional experiments to show that the C-terminus of MAP7 does not displace tau on its own (new Supplemental Figure 2f-g) indicating that MAP7 needs to be bound to the microtubule via its N-terminus to accumulate and displace tau over time.

3/ Did the author check that MAP7 and tau interact together? Indeed, the formation of a complex

between MAP7 and tau (for instance via the C-terminal part of MAP7) could also participate to the inhibition of tau binding to microtubules in the presence of MAP7.

We have performed additional experiments to show that MAP7 and tau do not physically interact with one another in solution biochemistry assays (new Supplemental Figure 2c). These results further suggest that MAP7 and tau may compete for overlapping binding sites on the MT.

4/ About the apparent K_D of MAP7 and tau for microtubules, why didn't the authors use the same tubulin concentrations for the two proteins (a smaller number of concentrations has been used to determine the binding curve of tau, Fig. 2j)?

Tau binding saturates at a higher tubulin concentration than MAP7, so we used a different range for each MAP. A similar amount of tau co-pellets at 1uM tubulin (39%) compared to the amount of MAP7 that co-pellets at 0.25uM tubulin (38%), so we set these as the starting concentrations for each MAP. However, while MAP7 appears to saturate at 4 uM, tau does not saturate until 8 uM tubulin in our assay conditions.

The apparent K_D of tau for microtubules seems higher compared to the one previously determined in the literature by similar assays holding tau constant and varying tubulin concentrations (about 0.1-0.3 μM , see for instance Duan and Goodson, 2012, Mol Biol Cell, 23: 4796-806; Elie et al, 2015, Sci Rep, 5:9964). The fact that tau binds microtubules before MAP7 (Fig. 2e) also suggests a higher affinity of tau for microtubules compared to MAP7, and thus a lower K_D value.

How do the authors explain these differences?

Duan and Goodson (2012) state: "*we found that the apparent affinity of Tau for MTs depends both on the approach used to study the interaction and on the concentration of each binding partner. We found that this dependence is observed when the data are fitted using both standard binding models or binding models designed to account for Tau-Tau oligomerization at the MT surface, suggesting that an interaction not accounted for by these models is occurring in these reactions.*"

There are therefore a number of reasons to explain the differences in the reported K_D of tau for microtubules. For their buffer, Duan and Goodson (2012) used 100mM Pipes with no additional salt compared to our buffer, which contains 150mM K-acetate. We used this salt concentration for all of our TIRF and solution biochemistry assays. Thus, the higher ionic strength of our buffer could be a contributing factor to the higher K_D for tau that we observe. They also used 1uM tau compared to our concentration of 500nM, which could potentially produce a different K_D due to the potential cooperativity of tau molecules.

Elie et al. (2015) used a buffer with 50mM KCl, but if you look closely at their co-pelleting graph, they were only able to pellet 50-60% of the tau at their highest tubulin concentration (4uM), so unfortunately, because they did not reach a saturating concentration of tubulin, an accurate K_D cannot be derived from this data. However, the amount of tau that does pellet at their highest concentration (4uM) is similar to our data.

The observation that tau binds the microtubule first is interesting, and we point this out in the text. We also note that tau has a ~40-fold lower residency time (τ) than MAP7. Since $K_D = k_{\text{off}}/k_{\text{on}}$ and $k_{\text{off}} = 1/\tau$, then $K_D = 1/(\tau)(k_{\text{on}})$. Therefore, the K_D relies both on the on-rate and the residency time of the protein. Thus, even if the K_{on} for tau is higher than MAP7, its lower residency time could result in a higher K_D for tau compared to MAP7. Unfortunately, our current imaging conditions limit our ability to accurately

measure the rapid k_{on} for individual tau molecules, but this will be an interesting line of future investigation.

5/ The authors reconstitute the behaviour of kinesin-1 along microtubules in the presence of both MAP7 and tau (Supplementary Fig. 3 and Fig. 4k). They show that MAP7 (50 nM) does not change the behaviour of K560 (10 nM) when encountering a patch of tau (100 nM) bound to microtubules, although MAP7 is still able in these conditions to recruit K560 outside the tau patches (Fig. 4k). What happens if the authors increase the amount of MAP7? By varying MAP7 concentrations, tau should progressively detach from the microtubules, restoring K560 processivity along microtubules. Such an experiment would nicely recapitulate part of the model proposed by the authors in Fig. 5h.

We thank the reviewer for proposing this exciting experiment. We performed this experiment and observe K560 being progressively recruited to areas in which it was initially excluded by tau as MAP7 invades and displaces tau patches (new Figure 4k).

Minor points:

- Regarding the single molecule analysis (Fig. 2k), it is very difficult to see on the graph the cumulative frequency of tau dwell times. The author should present the data obtained for tau and MAP7 on two separate graphs with an appropriate scale for the dwell-time axis.

We have added the graph of tau alone to new Supplemental Figure 2h.

- The significance of differences between some data sets is lacking on several graphs (e.g. Fig. 4 h and j, Fig. 5b and c)

We have now added stars to denote statistical significance for these graphs and others.

Reviewer #1 (Remarks to the Author):

The authors have adequately addressed concerns raised in a previous round of review. The manuscript is now acceptable for publication.

Reviewer #2 (Remarks to the Author):

As in my original review, I believe that the in vivo data are correlative and indirect. Their inclusion does not add much to the message of the paper, as they do not study the transport but use da neuron morphology instead. r

However, obviously, the authors believe that they need include the da neuron results in this manuscript and I do not think that it should prevent publication of the manuscript. I am happy with other revisions.

Reviewer #3 (Remarks to the Author):

The authors have addressed my comments and I recommend to accept this paper for publication.

Revision #2: Summary of Changes to Monroy *et al.* Manuscript

Changes to the manuscript and figures

We have made changes (in track-changes) to address the manuscript editing points in the email from the editor. We have shortened all figure legends to 350 words, but due to this, we had to split Figure 4 into two figures, making a total of 6 figures. We have not added or removed any data.

Response to Reviews

Reviewer #1 (Remarks to the Author):

The authors have adequately addressed concerns raised in a previous round of review. The manuscript is now acceptable for publication.

We thank this reviewer for their helpful comments.

Reviewer #2 (Remarks to the Author):

As in my original review, I believe that the *in vivo* data are correlative and indirect. Their inclusion does not add much to the message of the paper, as they do not study the transport but use *da* neuron morphology instead. r

However, obviously, the authors believe that they need include the *da* neuron results in this manuscript and I do not think that it should prevent publication of the manuscript. I am happy with other revisions.

We thank this reviewer for their helpful comments. We feel it is important to keep the *in vivo* work in the paper, as it shows the opposite effects of tau and MAP7/*ensconsin* overexpression in the same neuronal system and the effects of MAP7/*ensconsin* on kinesin transport in a neuronal system. We appreciate that this reviewer can accept the paper without removing the *in vivo* results.

Reviewer #3 (Remarks to the Author):

The authors have addressed my comments and I recommend to accept this paper for publication.

We thank this reviewer for their helpful comments.

Revision #1: Summary of Changes to Monroy *et al.* Manuscript

Changes to the manuscript

We have made changes throughout the text to reflect our new data (as described below), and in response to comments from the reviewers.

Changes to the figures

In response to the valuable suggestions by the reviewers, we have added new data to provide additional support for our models.

Figures 1, 2, and 3 These figures remain unchanged.

Figure 4	Stars indicating significance have been added to panels h , j , and l . Panel k has been replaced with data from the experiment that was suggested by reviewer 3. Panel k shows that although patches of tau exclude K560 from the microtubule, over a 10 minute period, the presence of MAP7 displaces tau from the microtubule and recruits K560 to the lattice, providing data for the model in Figure 5.
Figure 5	Stars indicating significance have been added to panels b , c , and f .
Supplementary Figure 1	This figure remains unchanged.
Supplementary Figure 2	Panel a is unchanged. New panel b shows the saturation curve for MAP7 Δ C compared to full-length MAP7 in panel a . New panel c shows a pull-down between purified tau and MAP7 to reveal that these two proteins do not interact. New panels d-e show that MAP7 Δ N does not bind to microtubules in TIRF-M (d) or microtubule co-pelleting (e) assays. New panels f-g show that MAP7 Δ N does not displace tau from the microtubule lattice. New panel h is the graph of tau residency times on its own, which is a zoom-in of the graph from Figure 2k.
Supplementary Figure 3	Panels a-e correspond to panels b-f from the original Supplementary Figure 2.
Supplementary Figure 4	This figure corresponds to the original Supplementary Figure 3 and none of the panels have been modified.

Point-by-Point Responses to Reviewers

Reviewer #1 (Remarks to the Author):

This interesting paper examines the roles of structural microtubule (MT)-associated proteins (MAPs) in regulation of MT-based transport. The authors investigate how binding of two common MAPs, tau and MAP7, to MT lattice influence motility of MT motor proteins in vitro and in vivo. These MAPs have been selected because of the known differences in their effects on motility of conventional MT motor protein kinesin-1. The authors show that MAP7 and tau compete for binding to MTs in vitro and that MAP7 can displace tau from MT lattice because of its higher MT binding activity, and longer MT residence time. The negatively charged C-terminal domain of MAP7 plays important role in the tau displacement. Overexpression tau and MAP7 homologue ensconsin in *Drosophila* larval dendritic arborization neurons have opposite effects on the numbers of dendritic branches or distribution of MT motor cargoes. These effects are consistent with inhibition of kinesin-1 motility by tau and stimulation by MAP7 in single molecule assays in vitro. MAP7 directly binds kinesin-1 end enhances its motility by facilitating interaction of this motor protein with MTs. The authors also show that binding to MTs of tau or MAP7 inhibits motility of another member of kinesin family kinesin-3 but has no effect on dynein/dynactin motility. Overall, this is important study that shows how competition among structural MAPs for binding to MTs can selectively affect motility of individual MT motor proteins and therefore MT-based transport in cells. The paper is clearly written, and experimental data are of high quality. This work will be likely of significant interest to broad scientific readership of Nature Communications. However, the paper can not be published as written and presented.

We thank the reviewer for their supportive comments about our work.

The authors indicate that full-size MAP7 removes tau from MTs but truncated MAP7 Δ C that lacks conserved negatively charged C-terminal region binds to MTs but does not remove tau. However, Fig.

2h shows that truncated MAP7 apparently displaced tau as well but at a significantly reduced rate (Fig. 2h).

After 10 minutes of imaging, tau intensity on the microtubule is reduced by 70% in the presence of full-length MAP7 compared to a reduction of 27% in the presence of MAP7 Δ C. We have now revised the manuscript for accuracy to state that removing the C-terminus of MAP7 greatly reduces its ability to evict tau from the microtubule (page 5).

The paper would benefit from further analysis of the mechanism underlying competition between the two MAPs. Whether C-terminal region of MAP7, similar to MAP7D3, has a separate MT binding site that enhances binding to MTs (a possibility that should be tested by determining KD for MAP7 Δ C) or direct interaction of MAP7 C-terminus with tau is responsible for the tau displacement?

We agree with the reviewer and have performed the suggested experiments to strengthen the mechanism of competition. We now show that the C-terminus of MAP7 does not bind microtubules on its own (new Supplemental Figure 2d-e). In fact, removing the C-terminus of MAP7 slightly increases the apparent binding affinity of MAP7 for microtubules (new Supplemental Figure 2b). The C-terminal half of MAP7 is highly negatively charged and may be repelled by the negatively charged C-terminal tails of tubulin. We also performed a pull-down between purified MAP7 and tau and have found that they do not interact with one another in solution (new Supplemental Figure 2c), suggesting this is not a mechanism for tau eviction. We thank the reviewer for suggesting these experiments and strengthening the paper.

The authors use truncated version of kinesin-1 K560 to study interaction with ensconsin and MAP7 and show that amino acid residues 500-523 of kinesin-1 are involved in this interaction. K560 construct terminates in the hinge-2 region and therefore lacks C-terminal tail domain involved in autoinhibition. However, a recent study (Barlan et al., 2013) indicates that kinesin-1 mutants without autoinhibitory function do not require ensconsin for transport and suggests that ensconsin activates kinesin-1 by relieving autoinhibition. Furthermore, pull-down experiments performed in this study failed to detect interaction between ensconsin and kinesin-1 in vitro. The authors should address discrepancy between their data and the results of Barlan et al. (2013).

Our pull-downs with purified proteins only suggest a modest interaction between MAP7/ensconsin with K560. We use the word “modest” in the text (page 7), because the interaction between MAP7 and K560 appears relatively weak in solution, these data may indicate a more transient interaction rather than a stable association of these two proteins. This could be why the pull-downs from lysates in Barlan et al. (2013) did not detect a robust interaction. Combined with our other data that indicate MAP7 does not tether K560 to the microtubule (Fig. 4f), our model is consistent with and expands upon the proposed model of Barlan et al. (2013), which we highlight on page 9.

It is possible that MAP7/ens could recruit as well as relieve the autoinhibition of full-length kinesin-1, and we have now added additional text in the discussion to address this point. It is very clear that MAP7/ens recruits non-auto-inhibited K560 to the microtubule, but it would be interesting in the future to test if MAP7/ens directly relieves the autoinhibition of kinesin-1.

Minor comments:

Panels b and c in legend to Supplemental Fig 2 are mislabeled.

Top right panel in fig. 4k seems to be mislabeled.

Typographical and grammatical errors should be corrected.

We thank the reviewer for catching these mistakes. We have rearranged a few figures and have thoroughly gone through the figures and manuscript to ensure proper labeling of figures and correction of grammatical errors.

Reviewer #2 (Remarks to the Author):

The paper by Monroy and colleagues describes competitive interactions between two well-known microtubule-binding proteins, tau and MAP7/ensconsin. The authors beautifully demonstrated that MAP7 promotes kinesin-1 interaction with microtubules, significantly stimulating microtubule-dependent ATPase activity of kinesin. Analysis of this effect using TIRF microscopy shows that the landing rate of kinesin on microtubules coated with MAP7 is significantly higher than on microtubules that do not contain it. Two-color TIRF data and co-sedimentation analysis show MAP7 has a higher affinity for microtubules than tau and displaces microtubule-bound tau, allowing kinesin to bind microtubules and initiate movement. Based on these beautiful in vitro data, the authors suggest a model that explains how kinesin is activated in axons containing tau and how inhibition of kinesin-3 by MAP7 explains its exclusion from the axons. I have no problem with in vitro results; these studies are well done and the data are novel and of a very high quality.

Unfortunately, the in vivo experiments do not quite match the quality of the biochemical work. First, static images of Golgi outpost distribution in DA neurons are used in the paper as a proxy for cargo transport. I can see several problems with this approach. First, It is not quite clear how outposts assume their steady-state distribution. It is possible that they are excluded from dendrites by kinesin-1, but the situation is complicated by the fact that not all microtubules are uniformly polarized in dendrites. Furthermore, as the authors know well, Golgi outposts themselves can nucleate microtubules which complicates interpretation of the results. Moreover, even if indeed ensconsin overexpression activates kinesin-1 in vivo, as the authors suggested, it is not a new piece of information, as ensconsin requirement for kinesin-1 activity in vivo has already been demonstrated in at least two publications. What makes this paper exciting is not MAP7-dependent kinesin requirement but a new concept of MAP7-tau competition that is so beautifully developed in the paper.

I want to emphasize once again, that biochemical data in the paper are novel and very interesting. These data alone are sufficient for a good publication, so my suggestion would be to publish biochemical results without the in vivo results presented in Fig. 3.

This is a nice story that should be published soon.

We thank the reviewer for their thorough and honest evaluation of the manuscript. We feel the inclusion of the in vivo data, while somewhat correlative and redundant with prior work, is still important for the overall message of the paper and supportive of the in vitro work presented. The microtubule orientation in the DA neurons is uniform in the primary branches (plus ends growing towards the cell body) and uniform in the terminal branches (plus ends growing towards the distal dendrite tips) (Stone et al., 2008; Ori-McKenney et al., 2012), thus the final placement of the Golgi outposts upon ensconsin overexpression is consistent with kinesin driving these cargo towards the location of plus ends. Even upon Golgi outpost nucleation of microtubules from the distal tips, for example, the enhancement of kinesin-1 activity by ensconsin overexpression would drive Golgi outposts and other cargo back towards the uniform tracks in the primary branches and eventually to the cell body, where the plus-ends are localized. We have modified our text to clarify a few of these points in support of the in vivo data.

Reviewer #3 (Remarks to the Author):

This manuscript addresses the relationship between MAP7 and tau activities on microtubules and consequences on the balance and distribution of microtubule motor proteins. Using biochemistry approaches and TIRF microscopy, the authors find that MAP7 and tau compete for microtubule binding in vitro and propose a mechanism by which MAP7 displaces tau from the lattice. They further show that MAP7 stimulates kinesin-based transport in vivo and enhances the interaction of kinesin-1 with microtubules in vitro, at the difference of the known inhibitory effect of tau on kinesin-1 motility. They also demonstrate that both MAP7 and tau inhibit the dendritic-specific motor kinesin-3 and that MAP7 has no effect on dynein in vitro. Based on their data, the authors propose a model explaining how the interplay between MAP7 and tau on the microtubule surface determines the distribution and activity of motor proteins. Overall I find this paper very interesting with data of high quality. It provides novel mechanistic insights into the regulation of motors by the combined activities of MAPs, a process that remains poorly understood.

However, they are some questions/clarifications that should be addressed prior to publication:

1/ Regarding the competition between MAP7 and tau, a truncated version of MAP7 containing the MTBD (MAP7 Δ C) is not able to displace tau from the microtubule surface (Fig. 2h). In that case, both MAP7 Δ C and tau can bind along the microtubule wall (Fig. 2g), which is not in favour of a same microtubule-binding site for the two proteins. How do the author explain this?

Although MAP7 Δ C does not displace tau to the same extent as full-length MAP7, the binding of MAP7 Δ C and tau is still mutually exclusive on the lattice, indicating they are still competing for overlapping binding sites on the microtubule. We have clarified this point in the text.

2/ Based on their results obtained with the mutant MAP7 Δ C, the authors propose that the C-terminal part of MAP7 plays a role in displacing tau molecules from microtubules. The authors should confirm this hypothesis by evaluating the specific effect of the C-terminal part of MAP7 on tau interaction along microtubules.

We have performed additional experiments to show that the C-terminus of MAP7 does not displace tau on its own (new Supplemental Figure 2f-g) indicating that MAP7 needs to be bound to the microtubule via its N-terminus to accumulate and displace tau over time.

3/ Did the author check that MAP7 and tau interact together? Indeed, the formation of a complex between MAP7 and tau (for instance via the C-terminal part of MAP7) could also participate to the inhibition of tau binding to microtubules in the presence of MAP7.

We have performed additional experiments to show that MAP7 and tau do not physically interact with one another in solution biochemistry assays (new Supplemental Figure 2c). These results further suggest that MAP7 and tau may compete for overlapping binding sites on the MT.

4/ About the apparent K_d of MAP7 and tau for microtubules, why didn't the authors use the same tubulin concentrations for the two proteins (a smaller number of concentrations has been used to determine the binding curve of tau, Fig. 2j)?

Tau binding saturates at a higher tubulin concentration than MAP7, so we used a different range for each MAP. A similar amount of tau co-pellets at 1 μ M tubulin (39%) compared to the amount of MAP7 that co-pellets at 0.25 μ M tubulin (38%), so we set these as the starting concentrations for each MAP.

However, while MAP7 appears to saturate at 4 μM , tau does not saturate until 8 μM tubulin in our assay conditions.

The apparent K_D of tau for microtubules seems higher compared to the one previously determined in the literature by similar assays holding tau constant and varying tubulin concentrations (about 0.1-0.3 μM , see for instance Duan and Goodson, 2012, Mol Biol Cell, 23: 4796-806; Elie et al, 2015, Sci Rep, 5:9964). The fact that tau binds microtubules before MAP7 (Fig. 2e) also suggests a higher affinity of tau for microtubules compared to MAP7, and thus a lower K_D value. How do the authors explain these differences?

Duan and Goodson (2012) state: “we found that the apparent affinity of Tau for MTs depends both on the approach used to study the interaction and on the concentration of each binding partner. We found that this dependence is observed when the data are fitted using both standard binding models or binding models designed to account for Tau–Tau oligomerization at the MT surface, suggesting that an interaction not accounted for by these models is occurring in these reactions.”

There are therefore a number of reasons to explain the differences in the reported K_D of tau for microtubules. For their buffer, Duan and Goodson (2012) used 100mM Pipes with no additional salt compared to our buffer, which contains 150mM K-acetate. We used this salt concentration for all of our TIRF and solution biochemistry assays. Thus, the higher ionic strength of our buffer could be a contributing factor to the higher K_D for tau that we observe. They also used 1 μM tau compared to our concentration of 500nM, which could potentially produce a different K_D due to the potential cooperativity of tau molecules.

Elie et al. (2015) used a buffer with 50mM KCl, but if you look closely at their co-pelleting graph, they were only able to pellet 50-60% of the tau at their highest tubulin concentration (4 μM), so unfortunately, because they did not reach a saturating concentration of tubulin, an accurate K_D cannot be derived from this data. However, the amount of tau that does pellet at their highest concentration (4 μM) is similar to our data.

The observation that tau binds the microtubule first is interesting, and we point this out in the text. We also note that tau has a ~40-fold lower residency time (τ) than MAP7. Since $K_D = k_{\text{off}}/k_{\text{on}}$ and $k_{\text{off}} = 1/\tau$, then $K_D = 1/(\tau)(k_{\text{on}})$. Therefore, the K_D relies both on the on-rate and the residency time of the protein. Thus, even if the K_{on} for tau is higher than MAP7, its lower residency time could result in a higher K_D for tau compared to MAP7. Unfortunately, our current imaging conditions limit our ability to accurately measure the rapid k_{on} for individual tau molecules, but this will be an interesting line of future investigation.

5/ The authors reconstitute the behaviour of kinesin-1 along microtubules in the presence of both MAP7 and tau (Supplementary Fig. 3 and Fig. 4k). They show that MAP7 (50 nM) does not change the behaviour of K560 (10 nM) when encountering a patch of tau (100 nM) bound to microtubules, although MAP7 is still able in these conditions to recruit K560 outside the tau patches (Fig. 4k). What happens if the authors increase the amount of MAP7? By varying MAP7 concentrations, tau should progressively detach from the microtubules, restoring K560 processivity along microtubules. Such an experiment would nicely recapitulate part of the model proposed by the authors in Fig. 5h.

We thank the reviewer for proposing this exciting experiment. We performed this experiment and observe K560 being progressively recruited to areas in which it was initially excluded by tau as MAP7 invades and displaces tau patches (new Figure 4k).

Minor points:

- Regarding the single molecule analysis (Fig. 2k), it is very difficult to see on the graph the cumulative frequency of tau dwell times. The author should present the data obtained for tau and MAP7 on two separate graphs with an appropriate scale for the dwell-time axis.

We have added the graph of tau alone to new Supplemental Figure 2h.

- The significance of differences between some data sets is lacking on several graphs (e.g. Fig. 4 h and j, Fig. 5b and c)

We have now added stars to denote statistical significance for these graphs and others.